# NINJ1 blocks HSV-1 entry into macrophages to impact viral replication and immunity

Ella Hartenian (ID), Magalie Agustoni & Petr Broz (ID) ✉

## Abstract

**Restriction factors block multiple stages of viral infection. Here we describe how Ninjurin1 (NINJ1) controls HSV-1 infection of macrophages, a key cell type that protects mice against infection. We observe that *Ninj1$^{-/-}$* mouse macrophages are more susceptible to HSV-1 infection than WT cells. Given the role of NINJ1 during cell death, we investigate whether its antiviral activity is linked to this function. Surprisingly, we do not observe differences in cell death at early timepoints post HSV-1 infection between genotypes. Instead, we attribute the higher infection rate of *Ninj1$^{-/-}$* macrophages to enhanced entry, with more viral particles entering each cell and a greater fraction of infected cells. The increased viral loads in *Ninj1$^{-/-}$* cells result in higher ISG and cytokine RNA expression, which we ascribe to both TLR signaling and STING-mediated recognition. Cytokine secretion, however, is severely dampened in infected *Ninj1$^{-/-}$* cells, pointing to greater viral replication suppressing the induction of inflammation. In conclusion, NINJ1 acts as a gatekeeper for HSV-1 entry in macrophages, impacting the inflammatory phenotype associated with HSV-1 infection.**

**Keywords** HSV-1; NINJ1; Cell Death; Virology; Restriction Factors
**Subject Categories** Immunology; Microbiology, Virology & Host Pathogen Interaction

## Introduction

Herpes simplex virus 1 (HSV-1) affects four in five people worldwide, making it one of the most ubiquitous viruses. Infections are lifelong, and symptoms vary between individuals, from asymptomatic to blisters, to severe outcomes, including blindness and encephalitis. HSV-1 initially infects keratinocytes and fibroblasts, then gains access to axons of the peripheral nervous system, where it establishes latency in sensory neurons and undergoes rounds of reactivation. When infection spreads beyond sensory neurons to the central nervous system (CNS), patients experience encephalitis—brain swelling and potentially debilitating symptoms.

Macrophages are innate immune sentinels and express many interferon-stimulated genes (ISGs), pattern recognition receptors

(PRRs) and cell death pathways components. While keratinocytes, fibroblasts and neurons are relevant cell types for HSV-1 infections, experiments in mice, especially in the brain, have consistently also shown a role for monocytic cells. Indeed, a recent single-cell analysis of an encephalitis model found that monocyte-like cells represented more than half of infected cells in the brain (Ding et al, 2025). Monocyte-like cells began infiltrating the brain 4 days post infection and peaked at 7 days post infection. A second study found that depleting macrophages from mice during a systemic intravenous infection or a vaginal infection resulted in decreased early control of the virus in the systemic case and death of the animals in both models (Lang et al, 2020). Furthermore, depleting microglia from mouse brains renders animals more susceptible to HSV-infection due to a dampened interferon (IFN) response (Katzilieris-Petras et al, 2022; Reinert et al, 2016). HSV-1 infection of human monocyte-derived macrophages also demonstrated cytokine production after infection (Melchjorsen et al, 2006). Together, these studies highlight the important role monocytic cells play in determining the outcome of infection.

One mechanism that host cells employ to limit viral replication is the induction of programmed cell death pathways, including pyroptosis, apoptosis and necroptosis. These pathways can act as sensors for infection. Suppression of apoptosis by viruses, on the other hand, engages necroptosis, which serves as a backup pathway of cell death in case apoptosis is inhibited (Kaiser et al, 2013). Previous reports have shown that HSV-1 modulates multiple cell death pathways during infection, including necroptosis, apoptosis and pyroptosis, inhibiting caspase-8, AIM2 and NLRP1-driven detection (Guo et al, 2018, 2022; Maruzuru et al, 2018; Parameswaran et al, 2024; Aubert and Blaho, 2001). Immune cells are also susceptible to cell death during infection. During a CNS model of mouse HSV-1 infection, microglia were shown to die by apoptosis, which had a beneficial effect on viral replication as it reduced the production of type-I IFN by these cells (Reinert et al, 2020).

Here, we describe a role for Ninjurin-1 (NINJ1), a plasma membrane resident protein highly expressed in the monocyte lineage of both human and mouse, in the restriction of HSV-1 infection. NINJ1 was originally described to be important for cell-to-cell adhesion (Lee et al, 2010), but has recently been shown to be essential for plasma membrane rupture and cellular lysis, the terminal event of multiple cell death pathways including apoptosis, ferroptosis, pyroptosis and necroptosis (Kayagaki et al, 2021; Ramos et al, 2024). NINJ1 undergoes self-oligomerization during cell death to form membrane-embedded filaments with a distinct

Department of Immunobiology, University of Lausanne, Lausanne 1066, Switzerland. ✉E-mail: petr.broz@unil.ch

hydrophilic and hydrophobic face. This feature allows NINJ1 filaments to create large pore-like lesions in the plasma membrane that release large cellular molecules such as danger-associated molecular patterns (DAMPs), chemokines, cytokines and nucleic acids (Kayagaki et al, 2021; Zhu et al, 2025). Most recently, NINJ1 has been shown to be a sensor of mechanical membrane stress, such that plasma membrane NINJ1 acts as "weak links" that break upon additional stress (Zhu et al, 2025), such as during necrosis-associated cell swelling (preprint:Hartenian et al, 2024). NINJ1 has further been shown to be important for the release of norovirus, a non-enveloped virus that both triggers and requires cell death for egress (Wang et al, 2023).

Here, we report that *Ninj1*-deficiency in mouse macrophages leads to enhanced infection with HSV-1. By monitoring HSV-1 GFP infection dynamics, we see earlier and stronger infection of *Ninj1*$^{-/-}$ macrophages as compared to WT controls. Although we observe initial differences in replication shortly after infection, *Ninj1*-deficiency has consequences for all stages of the HSV-1 lifecycle and eventually results in higher production of infectious viral particles. To understand how NINJ1 restricts HSV-1 infections, we first investigate if this is related to the role of NINJ1 in cell death. However, we observe that infected *Ninj1*-deficient cells do not die more or earlier than WT controls at early timepoints in infection, at which we already observe a difference in infection rates. Further exploring the effects of *Ninj1*-deficiency, we find that HSV-1 entry of *Ninj1*$^{-/-}$ macrophages is enhanced, with a higher infection rate and more viral particles entering each cell on average. Finally, we determine the consequences of the altered entry, measuring higher ISG RNA expression in infected *Ninj1*$^{-/-}$ cells, which we ascribe to both TLR signaling and STING-mediated recognition of nucleic acids. Cytokine protein production, on the other hand, is decreased in *Ninj1*$^{-/-}$ cells, which we attribute to the consequences of higher viral replication and thus higher levels of suppression. Together, this indicates that during infection of mouse macrophages, NINJ1 reduces HSV-1 entry, which alters the inflammatory potential of macrophages.

## Results and discussion

### NINJ1 controls the susceptibility of macrophages to HSV-1

NINJ1 is a membrane protein that functions in cell adhesion and cell death. As both processes are important for pathogens to successfully invade and replicate in host cells, we sought to determine if NINJ1 plays a role during infection with HSV-1. We therefore infected immortalized bone marrow-derived macro-phages (iBMDMs) from WT mice or mice deficient in *Ninj1*$^{-/-}$ with HSV-1 expressing a green fluorescence protein (HSV-1 GFP) under the control of the immediate early protein promoter ICP47 (see Reagents and Tools Table). Measuring the percentage of GFP-positive (GFP+) cells as a readout for infection, we found that *Ninj1*-deficiency resulted in twice as many infected cells compared to WT cells, as measured by flow cytometry at 6 h post infection (hpi) (Fig. 1A). Further analysis of infected cells showed that in the absence of NINJ1, more cells were infected and they also trended towards a higher mean fluorescent intensity (MFI), indicating that NINJ1 expression either resulted in dampened GFP expression or

altered early stages of infection (Figs. 1B and EV1A). To ensure that this effect was not an artifact due to the immortalization of the macrophages, we next examined infection of primary bone marrow-derived mouse macrophages (BMDMs) derived from WT or *Ninj1*$^{-/-}$ mice. Widefield microscopy showed an increase in GFP + cells in the absence of NINJ1 upon infection with HSV-1-GFP (Fig. 1C), and quantification by flow cytometry confirmed a threefold increase in the fraction of GFP+ *Ninj1*$^{-/-}$ cells as compared to WT (Fig. 1D), consistent with the results from immortalized BMDMs. To understand the kinetics of HSV-1 control by NINJ1, we analyzed the GFP signal intensity of infected cells every 10 m from the time of infection until 15 hpi. We observed that after an initial drop in the GFP signal, which occurred in both genotypes, there was a stronger increase in GFP intensity in *Ninj1*$^{-/-}$ cells as compared to the WT cells, starting around 4 hpi and plateauing around 11 hpi (Fig. 1E), consistent with our other measurements of HSV-1 GFP signal.

To confirm that the increase in HSV-1 GFP signal corresponded to an increase in viral gene products, we analyzed the expression of three viral proteins— ICP4 (immediate early), ICP8 (early), and gD (leaky late)—from 2.5 hpi to 8 hpi by immunoblotting in infected iBMDMs. Consistent with a stronger GFP signal of HSV-1 in the absence of NINJ1, *Ninj1*$^{-/-}$ macrophages not only showed higher expression levels of all three proteins, but also earlier expression of ICP4 and ICP8 (Fig. 1F). We next queried if the higher levels of viral protein expression resulted in the production of more infectious virions. We therefore collected supernatants from cells infected for 24 h and measured the number of infectious virions by plaque assay. We observed two logs higher titer from *Ninj1*$^{-/-}$ cells as compared to WT cells (Fig. 1G), consistent with a higher infection rate of *Ninj1*$^{-/-}$ cells.

To confirm that the observed phenotype was due to *Ninj1*-deficiency alone and not an off-target effect, we transduced *Ninj1*$^{-/-}$ iBMDMs with a vector expressing *Ninj1* under a doxycycline(dox)-inducible promoter. Treatment with dox induced expression of NINJ1 to levels comparable to WT cells (Fig. 1H). We next infected WT, *Ninj1*$^{-/-}$ and *Ninj1*$^{-/-}$ + *Ninj1* iBMDMs, in the presence or absence of dox, with HSV-1 GFP and measured the GFP signal over time. Complementing *Ninj1*-deficiency reduced the GFP signal to levels comparable to WT cells, confirming that *Ninj1*-deficiency was indeed the reason for increased viral replication (Fig. 1I). In the complemented cells in the absence of dox, the GFP signal followed the same trend as in *Ninj1*$^{-/-}$ cells, but was slightly attenuated, likely due to leaky expression of the construct (Fig. 1H). We also checked if overexpression of another, unrelated protein, could alter HSV-1 replication. Therefore, we lentivirally integrated a blasticidin resistance cassette into *Ninj1*$^{-/-}$ cells. After blasticidin selection, infection with HSV-1 GFP in the presence and absence of the blasticidin transgene resulted in similar levels of GFP expression (Fig. EV1B), demonstrating that overexpression of an unrelated protein does not alter HSV-1 replication. In summary, the results show that NINJ1 controls the susceptibility of mouse BMDMs to HSV-1.

### NINJ1 prevents infection in a cell-death-independent manner

Given the key role of NINJ1 in cell lysis downstream of multiple programmed cell death (PCD) pathways and the implication of

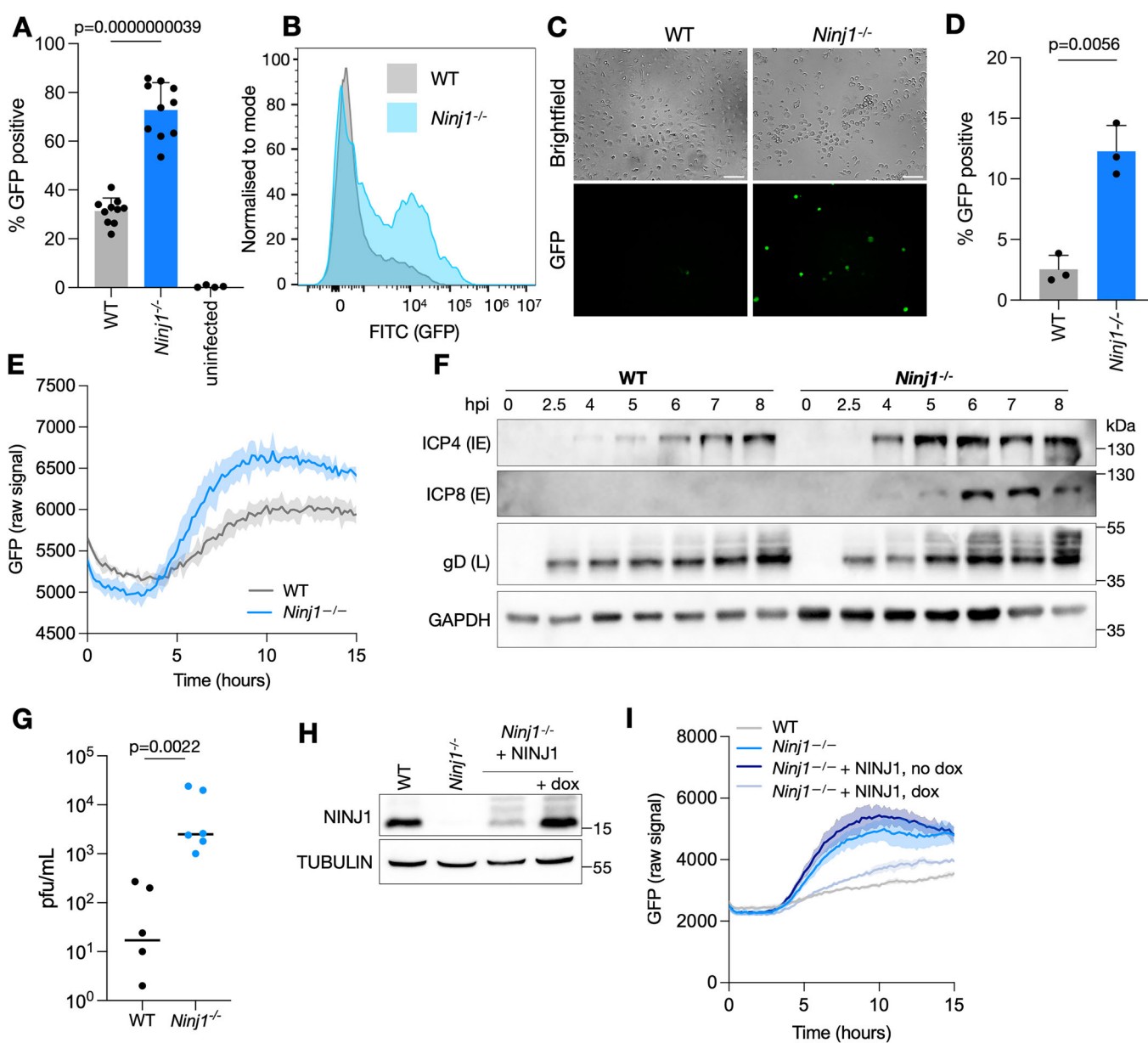

**Figure 1. NINJ1 controls the susceptibility of mouse macrophages to HSV-1.**

(**A**) WT and *Ninj1*⁻/⁻ iBMDMs were infected with HSV-1 GFP for 6 h at a MOI of 10 and subjected to flow cytometry to determine the percentage of infected cells. (**B**) WT and *Ninj1*⁻/⁻ iBMDMs were infected with HSV-1 GFP for 6 h at an MOI of 10 and subjected to flow cytometry, with the percentage of GFP-positive cells shown. (**C**) WT and *Ninj1*⁻/⁻ primary BMDMs were infected and visualized by microscopy after 6 h. Representative data from one experiment is shown (*n* = 5). (**D**) WT and *Ninj1*⁻/⁻ primary BMDMs were infected and subjected to flow cytometry with the percentage of GFP-positive cells shown. (**E**) WT and *Ninj1*⁻/⁻ iBMDMs were infected at a MOI of 10, and the GFP signal was measured every 10 m with a Citation5 Plate Reader. (**F**) WT and *Ninj1*⁻/⁻ iBMDMs were infected with HSV-1 GFP for the indicated timepoints, then subjected to immunoblotting for the indicated proteins. (**G**) Plaque assays from supernatant of WT and *Ninj1*⁻/⁻ iBMDMs infected with HSV-1 GFP for 24 h. (**H**) Immunoblot analysis of NINJ1 levels in WT and *Ninj1*⁻/⁻ iBMDMs complemented with a doxycycline (dox) inducible NINJ1 in the presence and absence of dox. TUBULIN serves as a loading control. (**I**) HSV-1 GFP signal measured every 10 m in WT, *Ninj1*⁻/⁻ iBMDMs and *Ninj1*⁻/⁻ iBMDMs complemented with a dox inducible NINJ1 ± dox with a Citation5 Plate Reader. Data information: In (**A**, **D**, **E**) data were presented as mean ± SD. Individual data points show biological replicates. A: unpaired *t*-test, *n* = 10, B: 1 representative experiment from 10 is shown, (**C**) one representative biological replicates of 3 is shown, (**D**) *n* = 3, unpaired *t*-test, (**E**) mean ± SD from 3 technical replicates is plotted, experiment is representative of four biological replicates, (**F**) 1 representative experiment of three biological replicates is shown, (**G**) dots are 1–2 technical replicates from three biological replicates with a line at the mean, Kolmogorov–Smirnov test, (**H**) one representative experiment of two biological replicates is shown, (**G**) mean ± SD from three technical replicates is plotted, experiment is representative of three biological replicates. Source data are available online for this figure.

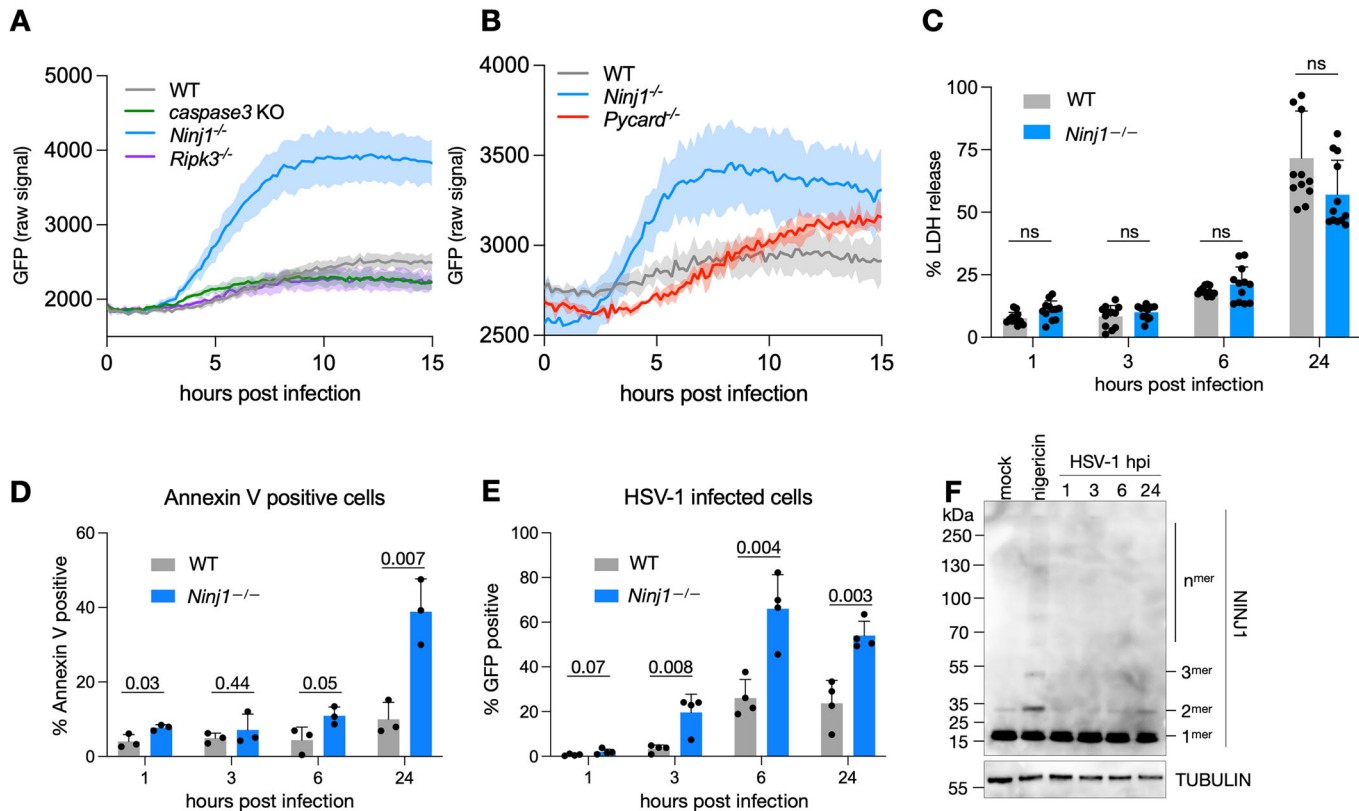

**Figure 2. NINJ1 prevents infection in a cell-death-independent manner.**

(A, B) The indicated genotypes of iBMDMs were infected with HSV-1 GFP and GFP signal was monitored every 10 m on a Citation5 plate Reader. (C) Lactate dehydrogenase (LDH) release was measured from the supernatant of HSV-1 GFP-infected WT and $Ninj1^{-/-}$ iBMDMs at the indicated hours post infection. (D, E) WT and $Ninj1^{-/-}$ iBMDMs infected with HSV-1 GFP were stained with Annexin V and subjected to flow cytometry, analyzing the Annexin V signal (D) and the percentage of infected cells (E). (F) Crosslinking immunoblot of WT primary BMDMs infected with HSV-1 GFP for the indicated periods of time. H3 serves as a loading control, and nigericin shows oligomerized NINJ1 upon cell death as a positive control. Data information: In (A, B) mean ± SD from three technical replicates is plotted, experiments are representative of three biological replicates. In (C–E), bars show the mean ± standard deviation. (C) Plotted dots are technical replicates from four biological replicates, unpaired t-test, ns not significant $n = 12$, (D, E) unpaired t-test $n = 3$–4, (F) one representative experiment of three biological replicates is shown. Source data are available online for this figure.

several of these pathways in the defense against HSV-1, we wished to determine if the inactivation of any PCD pathway phenocopied *Ninj1*-deficiency, which would implicate the respective pathway as a driver of NINJ1 activation. We therefore tested iBMDMs deficient for ASC (encoded by the gene *Pycard*), a key inflammasome adaptor protein needed for inflammasome-induced pyroptosis; RIPK3 (receptor interacting protein kinase 3), a critical necroptosis adaptor protein, and are unable to undergo necroptosis; and caspase-3, the apoptotic executioner caspase. We infected these cells in parallel with WT and $Ninj1^{-/-}$ iBMDMs with HSV-1 GFP and measured the GFP signal over time. While we observed a clear increase in the GFP signal in the $Ninj1^{-/-}$ cells, we did not detect a comparable increase in any of the other knock-out cell lines, suggesting that the individual removal of each of these cell death pathways did not phenocopy the $Ninj1^{-/-}$ cells (Fig. 2A,B).

We then asked if the absence of NINJ1 delayed macrophage death, allowing HSV-1 to replicate to higher levels. We therefore measured the release of lactate dehydrogenase (LDH), a marker for lytic cell death and NINJ1 activation, from HSV-1-infected WT and $Ninj1^{-/-}$ cells at 1, 3, 6, and 24 hpi. Even though *Ninj1*-deficiency had an impact on viral replication as early as 5–6 hpi (Fig. 1), we

did not observe any difference in LDH release between WT and $Ninj1^{-/-}$ macrophages at these timepoints (Fig. 2C). By 24 hpi, LDH release had increased to ~60% in both genotypes, with no significant differences between them.

Cell lysis is the terminal stage of lytic cell death pathways such as pyroptosis and necroptosis. Apoptosis, by contrast, does not induce cell lysis, and thus it is possible that WT and $Ninj1^{-/-}$ cells display different levels of apoptosis. We thus used Annexin-V staining, which is commonly used to detect the exposure of phosphatidyl serine (PS) on the plasma membrane. Annexin V staining showed low levels of PS exposure in both HSV-1-infected WT and $Ninj1^{-/-}$ cells at 1–6 hpi (Fig. 2d). At 24 hpi, the Annexin V signal increased significantly in the $Ninj1^{-/-}$ cells while remaining lower in the WT cells. The increase in the $Ninj1^{-/-}$ cells corresponded to a higher infection rate, which peaked in the $Ninj1^{-/-}$ cells at 6 h and declined slightly by 24 h (Fig. 2E).

A hallmark of NINJ1 activation during cell death is the oligomerization of the protein into filaments, which are required to induce cell lysis. Since our data suggest that NINJ1 plays a cell-lysis independent role during HSV-1 infections, we wished to confirm if NINJ1 oligomerized during HSV-1 infection by detecting

oligomeric NINJ1 by crosslinking immunoblots during infection. As a control, we included cells treated with nigericin, which induces pyroptotic cell death. Consistent with published reports (Kayagaki et al, 2021; Degen et al, 2023), we found that nigericin treatment resulted in the formation of NINJ1 dimers, trimers and higher-order oligomers, indicating NINJ1 activation (Fig. 2F). By contrast, we didn't observe any higher-order oligomers in HSV-1 infected cells. A low level of NINJ1 dimers was detectable, but this was comparable to untreated controls (Fig. 2F). This indicates that infection of WT macrophages with HSV-1 does not lead to oligomerization between 1 and 24 hpi, further confirming that NINJ1 must control HSV-1 infection independently of its role as an executioner of lytic cell death.

## Early stages of the viral lifecycle are impacted by the loss of NINJ1

Having ruled out an antiviral effect of NINJ1 related to its role in cell death, we next wished to distinguish at what stage of the viral lifecycle NINJ1 exerts its antiviral effect. Since we had observed a difference in GFP expression as early as 6 hpi (Fig. 1A,C,D) and a difference in ICP8 expression at 4 hpi (Fig. 1E), we speculated that NINJ1 modulated early events in the viral lifecycle. However, to rule out that *Ninj1*-deficiency affected post-viral gene replication events, we treated WT and *Ninj1*<sup>−/−</sup> cells with the viral DNA replication inhibitor Phosphonoacetic acid (PAA) during HSV-1 GFP infection and measured the GFP signal. PAA did not alter the higher infection rate of *Ninj1*<sup>−/−</sup> cells at 18 hpi (Fig. 3A). Western blotting for viral gene products at 18 hpi showed reduced expression of the leaky late protein gD in the presence of PAA (Fig. 3B), confirming the efficacy of PAA. Together, these results indicate that NINJ1 restricts the HSV-1 lifecycle at a stage prior to viral DNA replication.

## NINJ1 blocks viral entry into cells

Having determined that NINJ1 acts upstream of viral DNA replication, we examined the early steps of viral infection, starting with binding, the initial step of viral entry. To measure the number of viral particles bound to WT and *Ninj1*<sup>−/−</sup> cells, we incubated viral inoculum with cells at 4 °C, which allows viral binding but prevents entry, for 1 h, washed away unbound viral particles and compared the number of viral particles bound to cells by plaque assay. We observed slightly lower pfu/mL associated with *Ninj1*<sup>−/−</sup> cells as compared to WT (Fig. 3C), although the difference was not significant.

We next examined viral entry. HSV-1 can enter cells both by direct membrane fusion and by endocytosis, which is then followed by the fusion of the viral particle with the endocytic membrane. Endocytic entry has been proposed to be the major pathway for entry into macrophages (Lang et al, 2020). Right after endocytic uptake, there is a short window of time in which viral particles remain intact and infectious within endocytic vesicles. We therefore speculated that if NINJ1 restricts viral entry, more infectious virions could be recovered from endosomes in the absence of NINJ1. To assess the level of entry into cells, we thus infected WT and *Ninj1*<sup>−/−</sup> cells for 15 min (m) with HSV-1, treated cells with citric acid to inactivate all viral particles that had not entered cells and then determined the number of virions within endosomes by

plaque assay. We observed three times as many viral particles in *Ninj1*<sup>−/−</sup> cells as compared to WT cells (Fig. 3D), indicating that NINJ1 restricts HSV-1 entry into cells. To understand if this is a specific or general effect, we tested if *Ninj1*-deficiency impacted the endocytosis of fluorescent dextrans. We observed no change or even a slight decrease in endocytosis of *Ninj1*<sup>−/−</sup> macrophages (Fig. 3E). We then asked if entry of Vaccina virus (VACV), another enveloped virus that can enter by endocytosis was altered in the absence of NINJ1. We therefore infected WT and *Ninj1*<sup>−/−</sup> cells for 10 m with VACV and plaqued intact viral particles that had entered cells. Similar to HSV-1, we observed enhanced levels of VACV entry in *Ninj1*<sup>−/−</sup> cells compared to WT controls (Fig. 3F), suggesting that NINJ1 may have a general antiviral role by restricting viral entry through endocytic compartments.

We next used confocal microscopy to quantify the fraction of cells positive for ICP5, a HSV-1 capsid protein, at 45 m post infection. This method detects virions that enter by both direct membrane fusion and endocytosis. Compared to WT cells, more *Ninj1*<sup>−/−</sup> cells were found to be infected with HSV-1 (Fig. 3G,H). Moreover, counting the number of capsids inside each infected cell by microscopy showed that *Ninj1*<sup>−/−</sup> cells also tended to harbor more capsids per cell than WT cells (Fig. 3I). This confirms that HSV-1 can both enter more cells in the absence of NINJ1 and that the number of successful entry events per cell increases.

Finally, we asked if the absence of NINJ1 alters the expression of known HSV-1 entry mediators. We compared the presence of HVEM, Heparin Sulfate and Nectin-1 on the surface of WT and *Ninj1*<sup>−/−</sup> cells. There was slightly less Heparin Sulfate and HVEM detectable on the cell surface in *Ninj1*<sup>−/−</sup> cells as compared to WT cells, but this difference was too small to explain higher viral entry into these cells (Fig. 3J). Nectin-1 levels were not detectable on either genotype, with two different antibodies. In summary, these data demonstrate that NINJ1 restricts the entry of HSV-1 into macrophages, and this does not depend on altered expression of known HSV-1 entry mediators.

## NINJ1 is highly expressed in primary macrophages and in primary mouse tissues

Having characterized the role of NINJ1 as a gatekeeper for HSV-1 entry in macrophages, we wondered if it exerts the same role in other cell types. We thus examined NINJ1 expression across a panel of human and mouse cell lines and mouse tissue. We observed that the expression of NINJ1 in immortalized human cells lines was very low when compared to primary human monocyte-derived macrophages which had strong NINJ1 expression, consistent with previous reports (Borges et al, 2022) (Fig. 4A). Expression data from the human protein atlas also indicates that NINJ1 is most highly expressed in bone marrow lineage cells (Fig. 4B). In contrast, we detected NINJ1 expression across 11 different primary mouse tissues (Figs. 4C and EV2), indicating that NINJ1 is broadly expressed in mice. We also examined several mouse cell lines and found that NINJ1 was expressed in mouse embryonic fibroblasts (MEFs), RAW macrophages and 3T3 fibroblasts (Fig. 4D). To assess whether NINJ1 modulates HSV-1 infection in these cell types, as observed in macrophages, we infected CRISPR-generated NINJ1 knockout (KO) cells or KO controls with HSV-1. In MEFs, infection levels were comparable between NINJ1 KO and controls (Fig. 4E). In contrast, we observed elevated infection rates in NINJ1

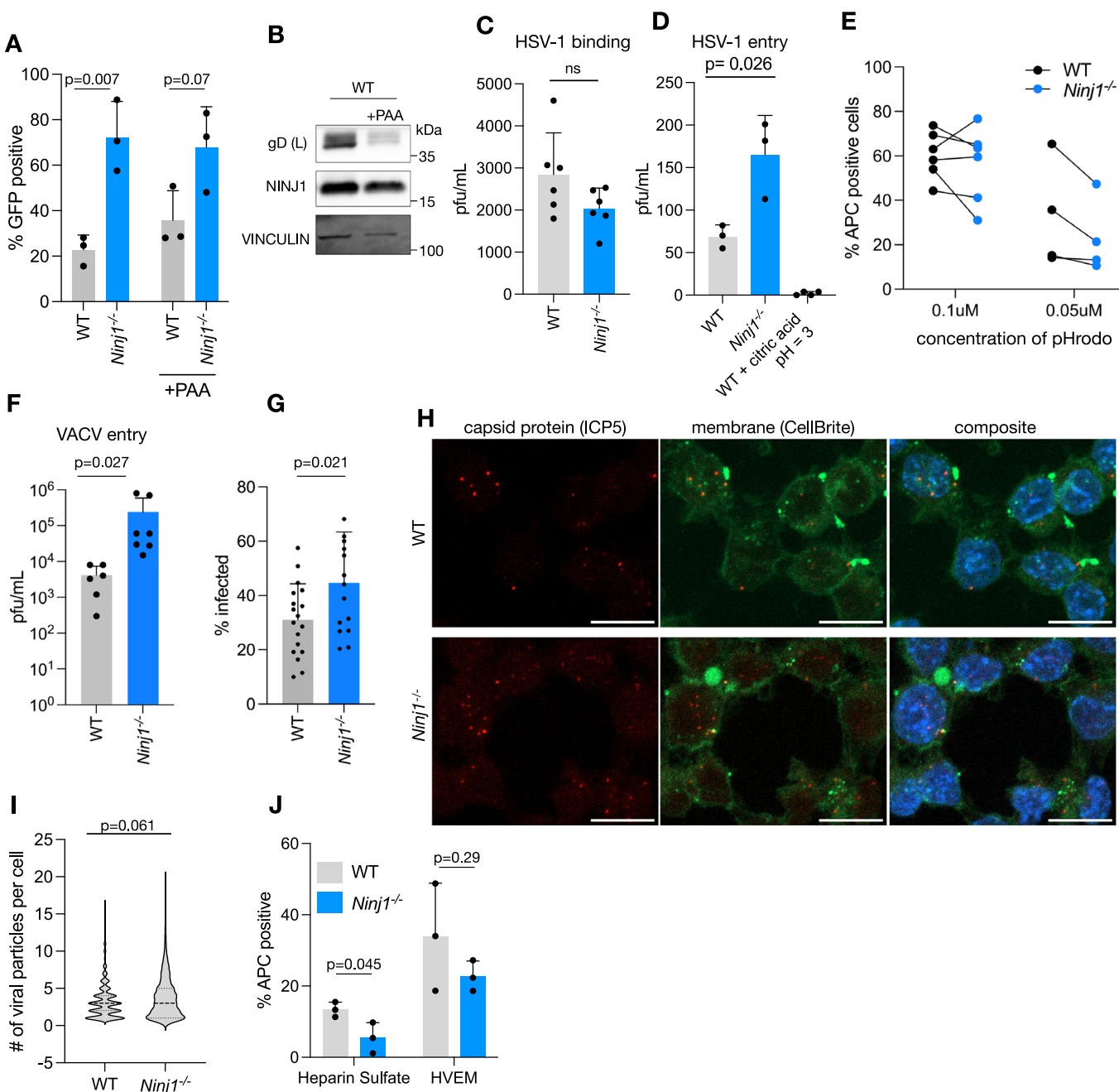

KO 3T3s compared to control cells (Fig. 4F), suggesting a role for NINJ1 in infection of these cells. In summary, these results demonstrate that mouse NINJ1 is widely expressed in both primary tissues and various cell lines, including RAW macrophages and fibroblasts, and that its impact on HSV-1 infection is cell-type-specific.

## HSV-1-induced cytokine secretion is suppressed in cells lacking NINJ1

Given our observation that more viral particles enter $Ninj1^{-/-}$ macrophages, we asked how this affects the innate immune response

to HSV-1. We therefore measured the levels of several cytokines and ISGs in WT and $Ninj1^{-/-}$ cells by RT-qPCR. We observed a trend of a stronger induction in $Ninj1^{-/-}$ cells with the most pronounced difference in $Viperin$ and $Ifn\beta$ expression and smaller differences in several other transcripts (Figs. 5A and EV3A). We thus asked what host pathway drives ISG induction in HSV-1-infected macrophages. Since TLRs and the cGAS/STING DNA sensing pathway have been implicated in the innate immune response to HSV-1 (Lima et al, 2010; Zhang et al, 2013; Reinert et al, 2016), we infected WT iBMDMs, $MyD88^{-/-}Trif^{-/-}$ cells, which abrogates signaling by TLRs, and $Sting^{-/-}$ cells, which is needed for cGAS-mediated responses. Comparing the induction of ISGs at 3 hpi and 6 hpi, we observed the most

◀ **Figure 3. Viral entry is enhanced in the absence of NINJ1.**

(A) WT and $Ninj1^{-/-}$ iBMDMs were infected for 18 h with HSV-1 GFP in the presence or absence of phosphonoacetic acid (PAA) and subjected to flow cytometry to measure the fraction of GFP+ cells. (B) Western blot of the leaky late protein gD, NINJ1, and VINCULIN in the presence and absence of PAA at 18 hpi. Representative of two biological replicates. (C) Plaque assay of HSV-1 GFP incubated with WT and $Ninj1^{-/-}$ iBMDMs for 1 h at 4 °C to measure viral binding. (D) Plaque assay of HSV-1 GFP incubated with WT and $Ninj1^{-/-}$ iBMDMs for 15 m and then washed with low-acid citric acid buffer to measure entry. Citric acid condition was HSV-1 and citric acid buffer added at the same time. (E) WT and $Ninj1^{-/-}$ iBMDMs were subjected to flow cytometry after incubation with the indicated concentration of fluorescent dextrans for 30 m to monitor endocytosis. (F) WT and $Ninj1^{-/-}$ iBMDMs were infected with VACV for 10 m, washed, and intracellular virions were plaqued. (G–I) Confocal fluorescent microscopy of cells infected with HSV-1 GFP for 45 m and stained with ICP5, cell brite and Hoechst. (G) Percentage of cells infected. (H) Representative images, scale bar 10 um. (I) Plot of the number of capsids (ICP5 positive foci) per WT and $Ninj1^{-/-}$ cell, defined using cellbrite stain. (J) WT and $Ninj1^{-/-}$ iBMDMs were subjected to flow cytometry after being incubated with Heparin Sulfate and HVEM primary antibodies. Data information: (A, D, F, J) mean ± SD are plotted with individual dots showing biological replicates, while (C) shows two technical replicates from three biological replicates. (A, D, F, G, J) unpaired t-test. (B) one biological replicate representative of 2, (E) dots show biological replicates with lines indicating those performed together, (G) each dot represents one of 14–17 fields of view taken from three biological replicates for a total of 750–950 cells, (I) violin plot with a solid line at the mean and dotted lines at quartiles, Kolmogorov–Smirnov test. Source data are available online for this figure.

marked decrease in $MyD88^{-/-}Trif^{-/-}$ cells (Fig. 5B). We also observed a decrease in $Sting^{-/-}$ cells, but this was more consistent for the induction of *Viperin* as compared to the induction of *IFI2712a*. To check if TLR and cGAS/STING signaling was intact in WT and $Ninj1^{-/-}$ cells, we stimulated WT and $Ninj1^{-/-}$ cells with the TLR ligand lipopolysaccharide (LPS) and cGAMP (2'3'-cyclic GMP-AMP), the activator of the cGAS/STING pathway. We observed similar induction of ISGs after stimulation with both ligands, confirming that both pathways were functional and comparable in the two cell lines (Fig. EV3B,C). In summary, we implicate both TLR-mediated sensing and cytosolic DNA sensing in the macrophage response to HSV-1, and show that accelerated replication of HSV-1 in $Ninj1$-deficient cells results in a faster immune detection.

Next, we asked if cytokine secretion followed the trend we observed in transcript levels of cytokines and ISGs. We thus performed a multiplex analysis of supernatants from HSV-1-infected WT and $Ninj1^{-/-}$ macrophages. In striking contrast to the elevated transcript levels, we observed that $Ninj1^{-/-}$ cells secreted three to six times less cytokines than WT cells (Fig. 5C). This difference was detectable for five out of six analyzed cytokines, namely for IL-6, IL-10, TNFα, IL-12 and IL-1β. To determine if this phenotype was specific to HSV-1 infection or if $Ninj1$-deficient cells had an intrinsic defect in cytokine secretion, we stimulated WT and $Ninj1^{-/-}$ cells with LPS for 2, 4, and 6 h and measured the levels of TNFα in the supernatant by ELISA. We observed no defect in TNFα secretion at any timepoint in the $Ninj1^{-/-}$ cells. Indeed, at 6 h post LPS stimulation, TNFα release was even slightly higher in $Ninj1^{-/-}$ cells than in WT cells (Fig. EV4A). We thus reasoned that higher levels of HSV-1 gene expression could block secretion of cytokines in $Ninj1^{-/-}$ cells, either by preventing their production or their release. To distinguish between these possibilities, we examined intracellular levels of IL-6 during HSV-1 infection, a cytokine that was detected to high levels in our multiplex analysis. We observed no defect in intracellular levels of IL-6 in the $Ninj1^{-/-}$ cells despite detecting more infected cells (Fig. EV4B,C), suggesting that increased levels of viral replication prevent the secretion, rather than the production, of cytokines.

In summary, we report that NINJ1 acts as a gatekeeper for HSV-1 entry in mouse macrophages. We see that in the absence of NINJ1, a higher proportion of cells are infected with HSV-1, resulting in the production of more infectious virions. We specifically pinpoint the difference in infection rates to a difference in viral entry, with more

viral particles entering $Ninj1^{-/-}$ cells and a greater percentage of $Ninj1^{-/-}$ cells getting infected. Thus, while NINJ1 has recently been identified as a key mediator of membrane rupture and cell lysis during cell death, we observe a novel, cell death-independent role for NINJ1 that renders macrophages refractory to HSV-1 infection at the site of first virus-cell contact. This work is in line with an intriguing novel hypothesis in the innate immune field, suggesting that the first line of defense against viral infection is PRR-independent and controls infection in the absence of generating an inflammatory response (Paludan et al, 2024).

Macrophages are recruited to the site of infection during both systemic (IV) and cutaneous mouse models of infection. During an encephalitis model, monocyte-like cells represented more than half of the total population harboring viral transcripts, indicating that HSV-1 enters these cells and that viral mRNA is produced (Ding et al, 2024). Mouse models of HSV-1 infection that lack macrophages succumb to disease much more quickly than controls, and microglia significantly attenuate the severity of CNS models of infection, highlighting the key role macrophages play in controlling lytic infection (Lang et al, 2020; Katzilieris-Petras et al, 2022). NINJ1 is highly expressed in macrophages and bone marrow-derived cells in comparison to most other cell types; however, the specific expression levels of NINJ1 in microglia of the brain has not been established. Furthermore, future work is required to understand the role of macrophages during human infection, the natural host of HSV-1. Here we observe that in the absence of NINJ1, ISG transcript expression increases, dependent on both TLR and cytosolic nucleic acid-dependent pathways. Surprisingly, despite transcript levels, infected $Ninj1^{-/-}$ cells release less cytokines than WT cells. While this defect could be due to a low level of pore-forming activity of NINJ1 that we are unable to detect by crosslinking immunoblot, we disfavor this hypothesis as the examined cytokines are released by the canonical ER-Golgi secretion pathway (Stow et al, 2009). This distinguishes them from IL-1 family cytokines, including IL-1β, that are released through Gasdermin pores (Xia et al, 2021). In addition, stimulating cells with LPS showed comparable release of TNFα in the presence and absence of NINJ1, indicating no intrinsic defect in $Ninj1^{-/-}$ cells for cytokine release. We thus favor a hypothesis where the loss of NINJ1 leads to more HSV-1 entry and increased viral gene expression that dampens cytokine secretion. Previous reports have found that HSV-1 suppresses pro-inflammatory cytokine

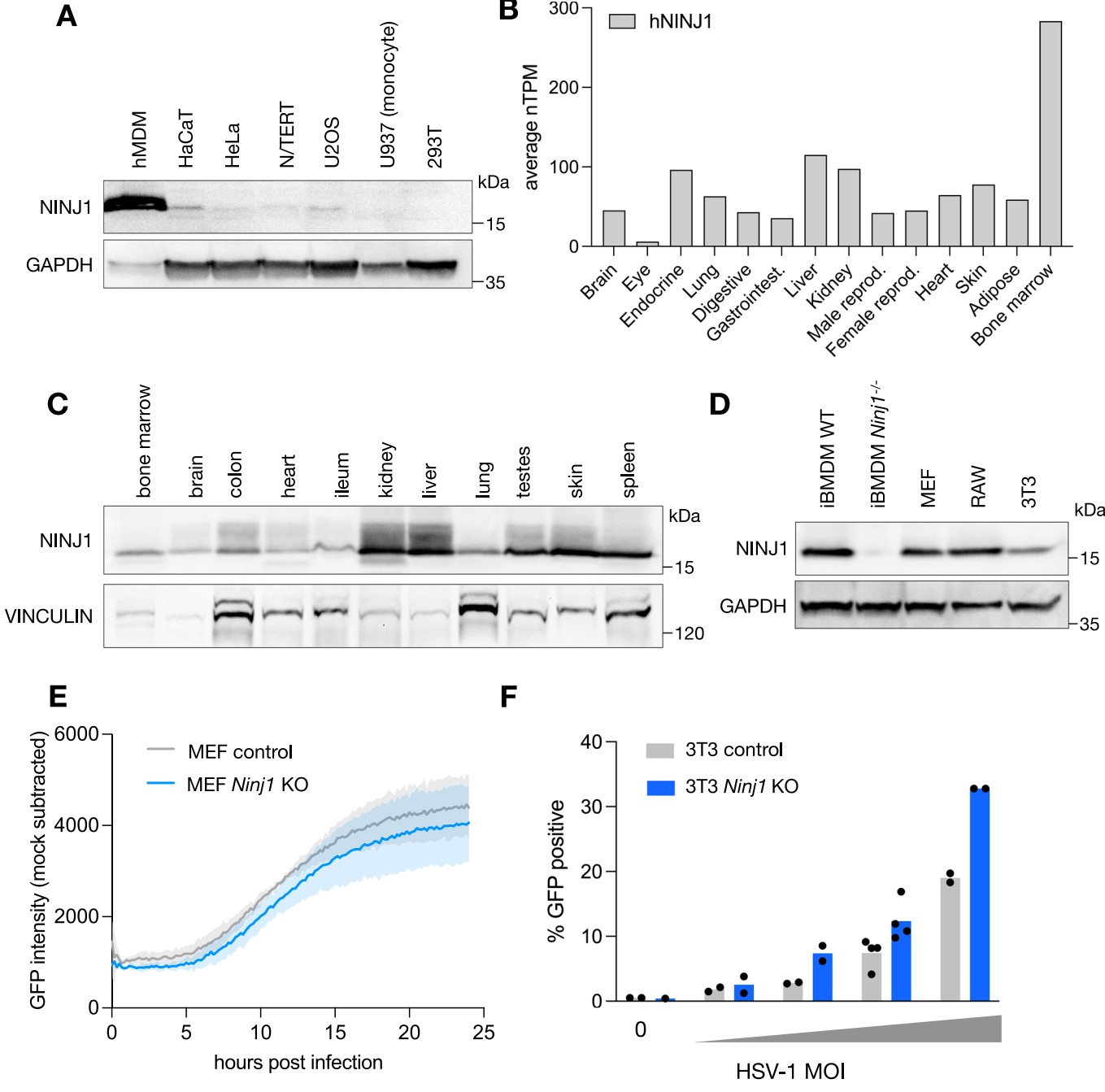

Figure 4. NINJ1 is highly expressed in primary macrophages and mouse tissue.

(A) The indicated cell lines were subjected to immunoblot for human NINJ1 and GAPDH as a control. (B) NINJ1 transcripts per million across different human tissues. Data from the Human Protein Atlas. (C) Organ homogenates from a WT mouse were blotted for NINJ1 and VINCULIN as a loading control. (D) The indicated cell lines were subjected to an immunoblot for mouse NINJ1 and GAPDH as a control. (E) Infection of control (sgluciferase) and *Ninj1* KO MEFs with HSV-1 GFP was measured by detecting the GFP signal every 10 m on a Citation5 Plate Reader. (F) Infection rate of control (sgluciferase) and *Ninj1* KO 3T3s with increasing amounts of HSV-1 GFP for 8 h as measured by flow cytometry. Data information: (A, D) are representative of three biological replicates. (C) is representative of two mice, see Fig. EV2 for additional data. (E) Line is mean ± SD from three technical replicates, representative of three biological replicates, (F) bar is the mean with dots indicating biological replicates $n = 2$-4. Source data are available online for this figure.

expression, although these reports detected defects in cytokine RNA levels (Mogensen et al, 2004)

At the resting state, NINJ1 is known to exist as autoinhibited dimers (Pourmal et al, 2025), which we also observe. NINJ1 has

further been described to maintain plasma membrane stability (Zhu et al, 2025). Thus, NINJ1 dimers at the resting state may change biophysical properties of the membrane, like fluidity, that have consequences for viral entry. In addition to being present on

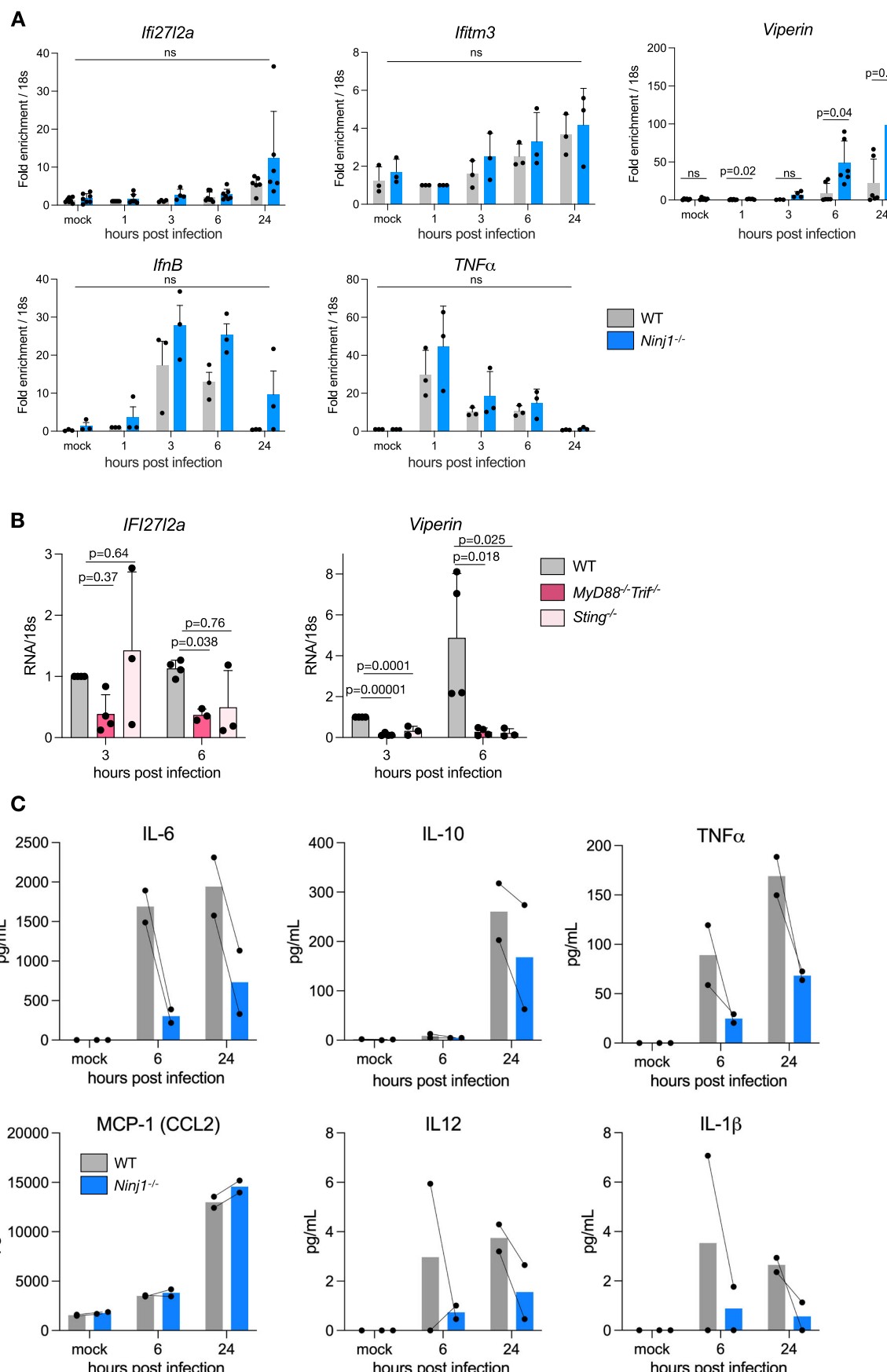

◄ **Figure 5. Loss of NINJ1 results in lower cytokine secretion.**

(A) WT and *Ninj1*⁻/⁻ iBMDMs were infected for the indicated times with HSV-1 GFP, and RNA levels of the indicated transcripts were measured by RT-qPCR. (B) WT, *MyD88*⁻/⁻*Trif*⁻/⁻, and *Sting*⁻/⁻ iBMDMs were infected with HSV-1 GFP for the indicated times, and RNA levels of the indicated transcripts were measured by RT-qPCR. (C) WT and *Ninj1*⁻/⁻ iBMDMs were infected for the indicated times with HSV-1 GFP, and supernatants were analyzed for the indicated cytokines. Data information: (A, B) mean ± SD is plotted with dots from biological replicates, (A) paired *t*-tests, ns not significant, *n* = 3–6. (B) one-way ANOVA with Dunnett's multiple comparison test, *n* = 3–4, (C) bars show the mean and each dot represents a biological replicate (*n* = 2), lines show samples from experiments performed together. Source data are available online for this figure.

the plasma membrane, NINJ1 is also found on the nuclear membrane and in the secretory system (likely during trafficking to the plasma membrane)(Degen et al, 2023), and we cannot rule out secondary roles for NINJ1 during infection elsewhere in the cell. Furthermore, we do not observe oligomerization of NINJ1 during HSV-1 infection, even though we observe macrophages releasing LDH at late stages of infection, indicative of lytic cell death. While HSV-1 is known to be able to trigger apoptosis, necroptosis and pyroptosis (Aubert and Blaho, 2001; Guo et al, 2022; Parameswaran et al, 2024; Reinert et al, 2020), HSV-1 may directly block NINJ1 oligomerization. This could happen either by a direct viral protein-NINJ1 interaction or indirectly by viral-induced inhibition of cell death pathways that could then trigger NINJ1. Alternatively, LDH release could occur in a NINJ1-independent manner at late stages of infection.

In conclusion, we demonstrate a novel role for NINJ1 in preventing entry of HSV-1 in mouse macrophages, a cell type that expresses the greatest amount of NINJ1 in the body and is essential for HSV-1 infection.

## Methods

### Reagents and tools table

| Reagent/resource | Reference or source | Identifier or catalog number |
|---|---|---|
| **Experimental models** | | |
| HEK 293T cells (*H. sapiens*) | ATCC | CRL 3216 |
| Vero cells (*C. aethiops*) | Florian Schmidt (Uni Bonn) | N/A |
| BSC-40 cells (*C. aethiops*) | Florian Schmidt (Uni Bonn) | N/A |
| Wildtype BMDM (*M. musculus*) | This study | N/A |
| *Ninj1*−/− BMDM (*M. musculus*) | This study | N/A |
| Wildtype iBMDM (*M. musculus*) | This study | N/A |
| *Ninj1*−/− iBMDM (*M. musculus*) | This study | N/A |
| MEF (*M. musculus*) | Dondelinger et al, (2023) | N/A |
| *Ninj1* ko MEF (*M. musculus*) | Dondelinger et al, (2023) | N/A |
| 3T3 (*M. musculus*) | Ramos et al, (2024) | N/A |
| *Ninj1* ko 3T3 (*M. musculus*) | Ramos et al, (2024) | N/A |

| Reagent/resource | Reference or source | Identifier or catalog number |
|---|---|---|
| L929 (*M. musculus*) | ATCC | NCTN Clone 929 |
| KOS HSV-1 ICP47 GFP | Florian Schmidt (Uni Bonn) | N/A |
| VACV Western Reserve EL EGFP | Florian Schmidt (Uni Bonn) | N/A |
| **Recombinant DNA** | | |
| pLVX mNINJ1 | Ramos et al, (2024) | Addgene 208775 |
| pLJM1 Blasticidin | This paper | N/A |
| psPAX2 | Addgene | 12260 |
| VSVG | Addgene | 12259 |
| **Antibodies** | | |
| Mouse anti-ICP4 | Santa Cruz | 56986 |
| Mouse anti-ICP5 | Abcam | 6508 |
| Mouse anti-ICP8 | Abcam | Ab20194 |
| Rabbit anti-mNINJ1 | Kayagaki et al, (2021) | N/A |
| Rabbit anti-hNINJ1 | This study | N/A |
| Mouse anti-GAPDH | Santa Cruz | 365062 |
| HRP-anti-TUBULIN | Abcam | Ab40742 |
| Rabbit anti-VINCULIN | Abcam | Ab91459 |
| HRP-conjugated goat anti-mouse | Southern Biotech | 1034-05 |
| HRP-conjugated goat anti-rabbit | Southern Biotech | 4030-05 |
| Alexa Fluor-568 goat anti-mouse | Invitrogen | A11031 |
| **Oligonucleotides and other sequence-based reagents** | | |
| Ifi27l2a forward | GCAATTGCCAATGGAGGTGG | N/A |
| Ifi27l2a reverse | CCTGCGTAGTCTGTACAGGC | N/A |
| Bst2 forward | CGAGACACAGGCAAACTCCT | N/A |
| Bst2 reverse | CCAGAGTTCACCTGCACTGT | N/A |
| Viperin forward | TTGGGCAAGCTTGTGAGATTC | N/A |
| Viperin reverse | TGAACCATCTCTCCTGGATAAGG | N/A |
| Ifitm3 forward | GGTGGGTGATGTGACTGGAG | N/A |
| Ifitm3 reverse | GAAGCACTTCAGGACCGGAA | N/A |
| IfnB forward | CACAGCCCTCTCCATCAACT | N/A |
| IfnB reverse | GCATCTTCTCCGTCATCTCC | N/A |
| TNFa forward | TCTTCTCATTCCTGCTTGTGG | N/A |
| TNFa reverse | GGTCTGGGCCATAGAACTGA | N/A |

| Reagent/resource | Reference or source | Identifier or catalog number |
| --- | --- | --- |
| Mx1 forward | ACCTGATGGCCTATCACCAG | N/A |
| Mx1 reverse | TTCAGGAGCCAGCTGTAGGT | N/A |
| Ifit2 forward | AGACAGTTACACAGCAGTCATGA | N/A |
| Ifit2 reverse | CACCCTGTCCTCAAACTCATC | N/A |
| Irf7 forward | CTGAAGTGAGGGGGGTCCAG | N/A |
| Irf7 reverse | CACAGCCCAGGCCTTGAAGA | N/A |
| 18 s forward | GTAACCCGTTGAACCCCATT | N/A |
| 18 s reverse | CCATCCAATCGGTAGTAGCG | N/A |
| **Chemicals, enzymes and other reagents** | | |
| DMEM | Gibco | 31966-021 |
| Fetal calf serum (FCS) | Bioconcept | 2-01F30-I |
| M-CSF | L929 cells | N/A |
| HEPES | Bioconcept | 5-31F00-H |
| NEAA | Gibco | 11140-035 |
| L-glutamine | Gibco | 25030-024 |
| Sodium pyruvate | Gibco | 11360070 |
| Pen/Strep | BioConcept | 4-01F00-H |
| DPBS | Gibco | 14190-094 |
| DPBS + CaCl$_2$ + MgCl$_2$ | Gibco | 14040-091 |
| Trypsin | BioConcept | 5-51F00-H |
| Opti-MEM | Gibco | 51985-026 |
| LT-1 | Mirus | MIR2300 |
| V-bottom 96-well analysis plate (TC treated) | Merck | P7116 |
| Cell culture microplate, 96-well (TC treated) | TPP | 92696 |
| Cell culture microplate, 24-well (TC treated | TPP | 92424 |
| Cell culture microplate, 12-well (TC treated) | TPP | 92412 |
| Cell culture microplate, six-well (TC treated) | TPP | 92406 |
| Cell culture microplate, six-well (non TC treated) | Costar | 2026-07-08 |
| Petri dish, 10 cm | Falcon | 351029 |
| T75 flask | TPP | 90076 |
| T150 flask | TPP | 90151 |
| LDH | Roche | 11644793001 |
| BSA | Merck | A9647- |
| PAA | Merck | 284270 |
| Carboxymethylcellulose | Merck | C5678 |
| Crystal Violet | Merck | V5265 |
| 4x Sample Buffer | Thermo Scientific | BN2003 |
| DTT | Thermo Scientific | R0862 |

| Reagent/resource | Reference or source | Identifier or catalog number |
| --- | --- | --- |
| Trans-Blot Turbo 5x Transfer Buffer | Bio-Rad | 10026938 |
| Protease inhibitors | Merck | 11836170001 |
| Precast gels 4–20% | Millipore | MP41G10 |
| Agarose | Merck | A9539 |
| Nitrocellulose Membranes | Amersham | 10600003 |
| NaCl | Merck | 1064045000 |
| KCl | Merck | P4504 |
| Citric acid | Merck | C0759 |
| CellBrite 488 | Biotium | BOT-3000-90-T |
| PFA | Electron Microscopy Science | 15710 |
| Saponin | Sigma | 47036 |
| Hoechst | Thermo Scientific | H3570 |
| Dextran, 10000 MW | Thermo Scientific | P10361 |
| Trizol | Thermo Scientific | 15596026 |
| Chloroform | Sigma Aldrich | C2432 |
| Glycogen | Thermo Scientific | AM9510 |
| Isopropanol | Merck | 67-63-0 |
| Ethanol | Merck | 64-17-5 |
| Methanol | Thermo Scientific | M/4000/17 |
| DNase kit | Thermo Scientific | AM1907 |
| MMLV | Life Technologies | 28025013 |
| RnaseOut RNase Inhibitor | Invitrogen | 10777-019 |
| SyberGreen Master Mix | Roche | 04887352001 |
| 2'3'-cGAMP | Invivogen | Tlrl-nacga23-02 |
| 0111:B4 LPS | Invivogen | Tlrl-3pelps |
| Mouse TNF-alpha DuoSet ELISA | R&D systems | Dy410 |
| APC mouse IL-6 | BioLegend | 504507 |
| Annexin V APC | BioLegend | 640920 |
| Annexin V Binding Buffer | BioLegend | 422201 |
| Annexin V Staining Buffer | BioLegend | 420201 |
| Permeabilization Wash buffer | BioLegend | 421002 |
| Blasticidin | Invivogen | Ant-bl-1 |
| Bs3 | Thermo Scientific | 21586 |
| SYBR Safe DNA Gel Stain | Invivogen | S33102 |
| 0.45um filter | Millipore | SLHPR33RS |
| **Software** | | |
| GraphPad Prism 10 | www.graphpad.com | |

| Reagent/resource | Reference or source | Identifier or catalog number |
|---|---|---|
| ImageJ | Imagej.nih.gov/index/html | |
| SnapGene | Snapgene.com | |
| FlowJo | Flowjo.com | |
| **Other** | | |
| LightCycler 480 | Roche | |
| CytoFlex S | Beckman Coulter | |
| Zeiss LSM800 confocal | Zeiss | |
| TissueLyser | Qiagen | |
| Transblot Turbo | Bio-Rad | |
| iBright | Thermo Scientific | |
| Cytation5 Imaging Reader | BioTek | |
| Epoch Plate Reader | BioTek | |

## Mammalian cell culture and generation

*WT* and *Ninj1⁻/⁻* mouse bone marrow-derived macrophages (BMDMs) were harvested and differentiated in Dulbecco's modified Eagle Medium (DMEM) (Gibco) containing 20% L929 supernatant as a source of macrophage colony-stimulating factor (M-CSF), 10% heat-inactivated fetal calf serum (FCS) (BioConcept), 10 mM Hepes (BioConcept), 1% penicillin/streptomycin, and 5 mL nonessential amino acids (NEAA, Gibco). Experiments were performed on days 9 to 10 of differentiation. Immortalization of macrophages was performed as previously described (Broz et al, 2010). Briefly, primary monocytes were extracted from bone marrow and then treated with filtered supernatant from existing iBMDMs for 24 h. Then these cells are grown for up to 2 months to wait for those that are immortalized to grow up. After several weeks in culture, M-CSF in the media is reduced from 20% to 10%. Immortalized macrophages (iBMDMs) were cultured in DMEM containing 10% FCS, 10% M-CSF, 10 mM Hepes, and 5 mL nonessential amino acids. HEK 293T and 3T3 cells were cultured in DMEM supplemented with 10% FCS. 3T3 KO and controls have been previously described (Ramos et al, 2024). MEFs were grown in DMEM with 10% FCS, 2 mM ʟ-glutamine, and 400 μM sodium pyruvate and were previously published (Dondelinger et al, 2023). All cells were grown at 37 °C, 5% $CO_2$. Cells are regularly tested for mycoplasma.

## Viral propagation, infection, and plaque assays

KOS HSV-1 GFP (ICP47 promoter drives GFP, a kind gift of Florian Schmidt) was propagated on Vero cells. Briefly, 90% confluent cells were infected at an MOI of 0.01 and left to grow for 3–4 days. Cell free viral particles were combined with cells, subjected to douncing, and spun at 500×g for 5 m. The supernatant was then ultracentrifuged at 15,000×g for 90 min, and the viral pellet was resuspended in DMEM with 10% FCS and 1 g/100 mL of BSA. Viral preps were titered on Vero cells, and MOIs reflect Vero cell numbers. Plaque assays were performed by seeding Vero cells at $1 \times 10^4$ cells per well in 24-well plates. The next day, cells were infected in 500 uL for 2 h, then

aspirated and replaced with full media supplemented with 0.5% carboxymethylcellulose, then left to grow for 3 days. Cells were fixed with 4% paraformaldehyde for 10 min, then aspirated and stained with 0.5% crystal violet for 10 m before counting.

Infections were performed at an MOI of 10 when not indicated otherwise. For macrophage infections, viral supernatants were left on cells for 3 h in opti-MEM before changing media, except for experiments analyzed by BioTek Cytation5 plate reader, where the virus was added to cells and not removed.

VACV (Western Reserve EL EGFP, a kind gift of Florian Schmidt) was propagated on BSC-40 cells. Briefly, two 90% confluent T150 flasks were infected at MOI 0.01 and 2 days later, scraped and subjected to two freeze-thaw cycles. Cellular debris was centrifuged out at 300×g for 10 min, and the supernatant was centrifuged at 15,000×g for 90 min. The viral pellet was resuspended in DMEM and titered on BSC-40 cells.

## Immunoblotting

For western blotting analysis, cells were lysed in 66 mM Tris-HCl, pH 7.4, 2% SDS, 10 mM DTT, and NuPage LDS sample buffer (Thermo Fisher). For organ samples, 30 mg of each organ was homogenized in 300 uL of RIPA buffer with protease inhibitors (Roche) in a TissueLyser (Qiagen) with a metal bead for 2 × 2 m at 30 Hz. Lysates were then spun at top speed for 3 min, and the supernatant was used for blotting. Cell lysates were separated on gradient precast gels 4–20% (Milipore) and transferred to a nitrocellulose membrane using Transblot Turbo (Bio-Rad). The antibodies used were: anti-ICP4 (Santa Cruz 56986), anti-ICP8 (Abcam 20194), anti-gD (Santa Cruz 69802) anti-mouse NINJ1 (rabbit IgG2b clone 25; a kind gift from Genentech; 1:8000), anti-human NINJ1 (commercially generated for this publication by Yenzym, 1:500), anti-tubulin (ab40742; Abcam; 1:2000), anti-GAPDH (365062; Santa cruz, 1:3000), anti-VINCULIN (Ab91459; Abcam; 1:1000). Primary antibodies were detected with horseradish peroxidase (HRP)-conjugated goat anti-rabbit (4030-05; Southern Biotech; 1:5000), HRP-conjugated goat anti-mouse (1034-05; Southern Biotech; 1:5000 or 12-349; MilliporeSigma; 1:2000) secondary antibodies.

## Complementation of NINJ1 in *Ninj1⁻/⁻* iBMDMs

pLVX mNINJ1 (addgene 208775) was made into a lentivirus by transfecting 3 million 293T cells in a six-well dish with 1.25 μg of packaging plasmid (pLVX mNINJ1), 1.25 μg of psPAX2 and 250 ng of VSVG and 8 uL LT-1 transfection reagent (Mirus). Media was changed 24 h post transfection to DMEM + 10% FCS + 1 g/100 mL bovine serum albumin (BSA), lentivirus was harvested 48 h post transfection and filtered through a 0.45 μm filter. 1.5 million iBMDMs were transduced with 1 mL of virus by spinning at 500×g for 1.5 h in one well of a non-tissue culture-treated six-well dish. Next, cells were split out into a 10 cm petri dish and selected with 1 μg/mL puromycin 24 h post transduction for at least 5 days. For transgene-expressing experiments, a pLJM1 vector carrying blasticidin resistance was packaged into lentivirus as described above for complementation and spinfected into *Ninj1⁻/⁻* cells as described above. Cells were then selected with 5 μg/mL Blasticidin for 5 days before experiments were performed, infecting cells ± the Blasticidin resistance cassette with the same volume of HSV-1.

## Entry experiments

For infections where entry by endocytosis was quantified, after addition of viral supernatants onto cells, cells were washed with a low pH citric acid buffer (135 mM NaCl, 10 mM KCl, 40 mM citric acid, pH 3) for 5 m to inactivate any extracellular viral particles.

## Annexin V flow cytometry

Macrophages were seeded in tissue culture-treated 12-well plates at $1 \times 10^5$ cells/well. Supernatant and adherent cells were combined by trypsinizing adherent cells and then staining with Annexin V APC (BioLegend) following the manufacturer's protocol, using BioLegend Cell Staining Buffer and Annexin V Binding Buffer (BioLegend). Cells were analyzed on a Cytoflex S. All flow cytometry analysis was performed in FloJo.

## Cell lysis assays

A day before stimulation, cells were seeded in 96-well plates at a density of $2 \times 10^4$ iBMDMs per well. Cells were infected at an MOI of 10. Cell lysis was quantified by measuring the lactate dehydrogenase (LDH) amount in the cell supernatant using the LDH cytotoxicity kit (Takara, Clontech) according to the manufacturer's instructions and expressed as a percentage of total LDH release. LDH release was normalized to untreated control and 100% lysis control, by adding Triton X-100 to a final concentration of 0.01% as follows: $(\text{LDH}_{sample} - \text{LDH}_{negative\ control}) / (\text{LDH}_{100\%\ lysis} - \text{LDH}_{negative\ control}) \times 100$.

## Crosslinking assay

WT BMDMs were seeded in 24-well plates at a density of $2 \times 10^5$ cells per well a day before stimulation. The next day, cells were infected with HSV-1 at an MOI of 50 for 3 h, or the relevant time post infection. Cells were washed once in $Ca^{2+}Mg^{2+}$ containing PBS, and then 200 uL of $Ca^{2+}Mg^{2+}$ containing PBS was left on the cells. Next, the crosslinker $BS^3$ (bis(sulfosuccinimidyl)suberate) was added according to the manufacturer's instructions. In brief, $BS^3$ was added to the media (3 mM; Thermo Fisher) and incubated for 5 m at room temperature. Next, a solution of 20 mM Tris pH 7.5 was added to stop the crosslinking reaction and incubated for 10 m at room temperature. Cell supernatants were collected, and proteins were precipitated with methanol and chloroform and combined with cell lysates for western blotting analysis.

## Confocal microscopy and image analysis

iBMDMs seeded onto glass coverslips were infected for 45 m at an MOI of 10. For the last 5 m of infection, they were treated with CellBrite (Biotium) to label the plasma membrane. Then cells were fixed in 4% PFA for 10 min, washed with PBS, permeabilized with 0.05% saponin and blocked with 1% BSA in PBS. Then, samples were incubated with an anti-ICP5 antibody (Abcam 6508), then an Alexa Fluor-568 conjugated antibody and Hoechst (1:1000). Samples were then imaged with a Zeiss LSM800 confocal laser scanning microscope using a 63x/1.4 NA oil objective. Quantification of capsids was performed using a maximum projection of the z-stack, by manually segmenting cells and counting the number of infected cells per field of view and the number of capsids per cell. All microscopy datasets were analyzed and processed using Fiji software. Raw microscopy images are available on BioImage Archive: https://doi.org/10.6019/S-BIAD2327.

## Dextran uptake assays

xiBMDMs were seeded at $5 \times 10^5$ in 100 uL of FACS buffer (PBS with 2% FCS and 3 mM EDTA) in a V-bottom 96-well plate. Fluorescent dextran was added to a final concentration of either 0.05 uM or 0.1 uM and incubated at 37 °C for 30 m with gentle agitation every 10 min. Each well was then supplemented with 100 uL FACS buffer, spun at 300×$g$ for 5 m at 4 °C, washed once with 200 uL FACS buffer, spun again as before and resuspended in 250 uL FACS buffer for analysis. After the initial 30 min incubation, samples were kept on ice to prevent additional dextran entry until being analyzed by flow cytometry.

## Antibody generation

Rabbit anti-human NINJ1 was made by YenZym antibodies, LLC to a synthetic NINJ1 peptide comprising the first 76 aa of human NINJ1 (GenScript).

## RT-qPCR

RNA was isolated with Trizol. Briefly, $1 \times 10^5$ cells seeded the day before and then infected were washed and resuspended in 50 uL of PBS, then lysed in 500 uL of Trizol. About 100 uL of chloroform was added and spun for 15 m at 21,000×$g$ at 4 °C. The top phase was moved into a new tube with 5 uL of glycogen (Invitrogen) and 500 uL of isopropanol. After 10 min at room temperature, samples were spun again for 10 m at 21,000×$g$ at 4 °C, and then washed twice with 70% ethanol, spinning for 5 min after each spin. Then ethanol was removed, samples were resuspended in Rnase/Dnase-free water. 1.5 ug of RNA was DNase-treated (Thermo DNA-free Kit) and then subjected to RT with MMLV following the manufacturer's guidelines. RT-qPCR was performed with Sybr-Green Master Mix on a LightCycler 480 (Roche). Primers used for RT-qPCR are listed in the Reagents and Tools Table.

## LPS and cGAMP stimulation

Cells were stimulated with 5 µg/mL 2'3' cGAMP (Invivogen) or 10 µg/mL 0111:B4 LPS (Invivogen) by adding the appropriate concentration to total media for the indicated amounts of time.

## Animals and primary cells

All experiments involving animals were performed under the guidelines and approval from the cantonal veterinary office of the canton of Vaud (Switzerland), license number VD3895. All mice were bred and housed at a specific-pathogen-free facility at 22 C° room temperature, 10% humidity and a day/night cycle of 12 h/12 h at the University of Lausanne. *Ninj1*-deficient mice have been described before (Degen et al). *Myd88*[-/-]*Sting*[-/-], *Sting*[-/-], *Ripk3*[-/-], *Pycard*[-/-] (ASC), and caspase-3 KO cells have been described before

(Heilig et al, 2020; Mariathasan et al, 2004). Animals used in the study were between 8 and 10 weeks old.

## Cytokine analysis

WT and *Ninj1⁻/⁻* immortalized BMDMs were infected for 3 h in opti-MEM, and then the media was changed to complete media. Infections were left for a total of 6 or 24 h, and then the supernatant was harvested and treated with 0.1% Triton X-100 to inactivate any viral particles. Eve Technologies (Calgary, Canada) performed the cytokine analysis shown in Fig. 5C.

## ELISA

Mouse TNF-alpha DuoSet ELISA (R&D systems DY410) was performed on 75 μL of supernatant (from a total of 200 μL) from 50,000 cells seeded at the time of stimulation in a 96-well plate. Each condition was performed in technical triplicate. ELISA was performed following the manufacturer's protocol.

## Intracellular IL-6 staining

Cells were seeded at 400,000 per well of a six-well dish and infected in Opti-MEM for 6 h. Cells were then scraped, fixed in 4% PFA for 10 m and permeabilized with BioLegend Intracellular Staining Permeabilization Wash Buffer, following the manufacturer's recommendations. APC Mouse IL-6 was then incubated for 1 h at RT, and samples were analyzed on a flow cytometer.

## Data availability

Microscopy data related to Fig. 3H from this study is available on BioImage Archive: https://www.ebi.ac.uk/biostudies/bioimages/studies/S-BIAD2327

The source data of this paper are collected in the following database record: biostudies:S-SCDT-10_1038-S44319-025-00638-8.

## Peer review information

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

## Acknowledgements

This work was supported by SNSF Project funding (310030_219286 and 310030B_192523) to P.B., EMBO postdoctoral fellowships ALTF 27-2022 to E.H. We would like to thank the UNIL animal facility for their help and support, Florian Schmidt for his kind gift of HSV-1 GFP, VACV, and various cell lines, Mathieu Bertrand (VIB) for his gift of control and NINJ1 KO MEFs, Vishva Dixit (Genentech) for sharing the mouse anti-NINJ1 antibody and the Broz lab for helpful discussions.

## Author contributions

**Ella Hartenian**: Conceptualization; Resources; Data curation; Formal analysis; Supervision; Funding acquisition; Validation; Investigation; Visualization; Methodology; Writing—original draft; Writing—review and editing. **Magalie Agustoni**: Validation; Investigation; Visualization; Methodology. **Petr Broz**: Conceptualization; Supervision; Funding acquisition; Writing—original draft; Project administration; Writing—review and editing.

Source data underlying figure panels in this paper may have individual authorship assigned. Where available, figure panel/source data authorship is listed in the following database record: biostudies:S-SCDT-10_1038-S44319-025-00638-8.

## Disclosure and competing interests statement

The authors declare no competing interests.

# Expanded View Figures

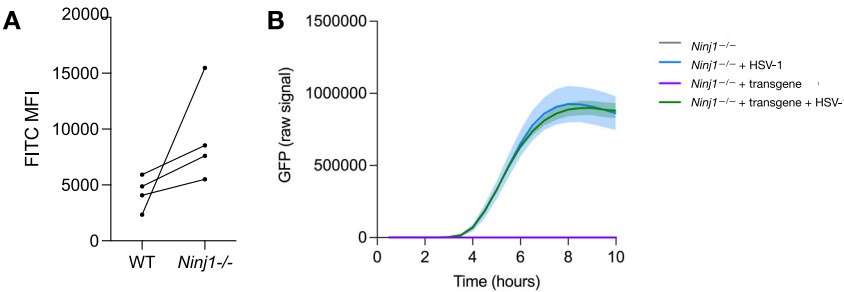

**Figure EV1.  NINJ1 controls the susceptibility of mouse macrophages to HSV-1.**

(A) WT and *Ninj1*⁻/⁻ iBMDMs were infected for 6 h with HSV-1 GFP, subjected to flow cytometry and the median fluorescence intensity was measured in FlowJo. (B) *Ninj1*⁻/⁻ iBMDMs ± integration of a blasticidin resistance transgene were infected with HSV-1 GFP, and GFP signal was measured every 10 min on a Citation5 Plate Reader. Data information: (A) dots are biological replicates with a line connecting data collected as part of the same experiment, (B) line is mean ± SD from three technical replicates, representative of two biological replicates.

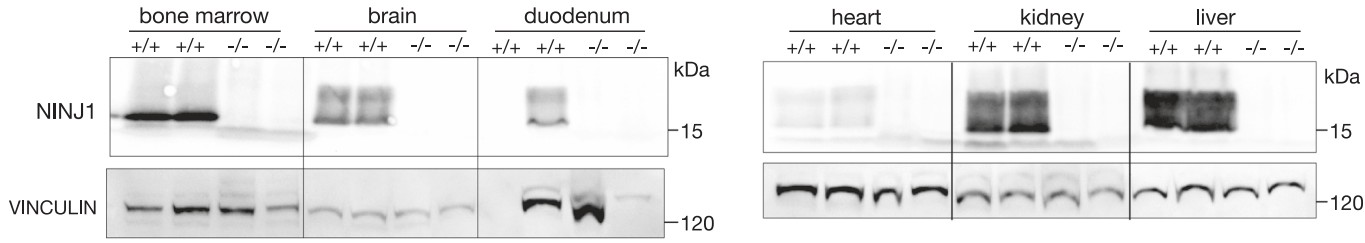

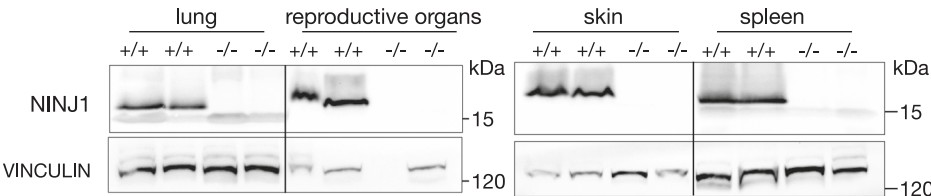

**Figure EV2. NINJ1 is highly expressed in primary macrophages and mouse tissue.**

Organ homogenates from two WT mice (+/+) and two *Ninj1⁻/⁻* mice (−/−), one male (left) and one female (right) of each genotype, were subjected to immunoblot for NINJ1 expression across the indicated tissues.

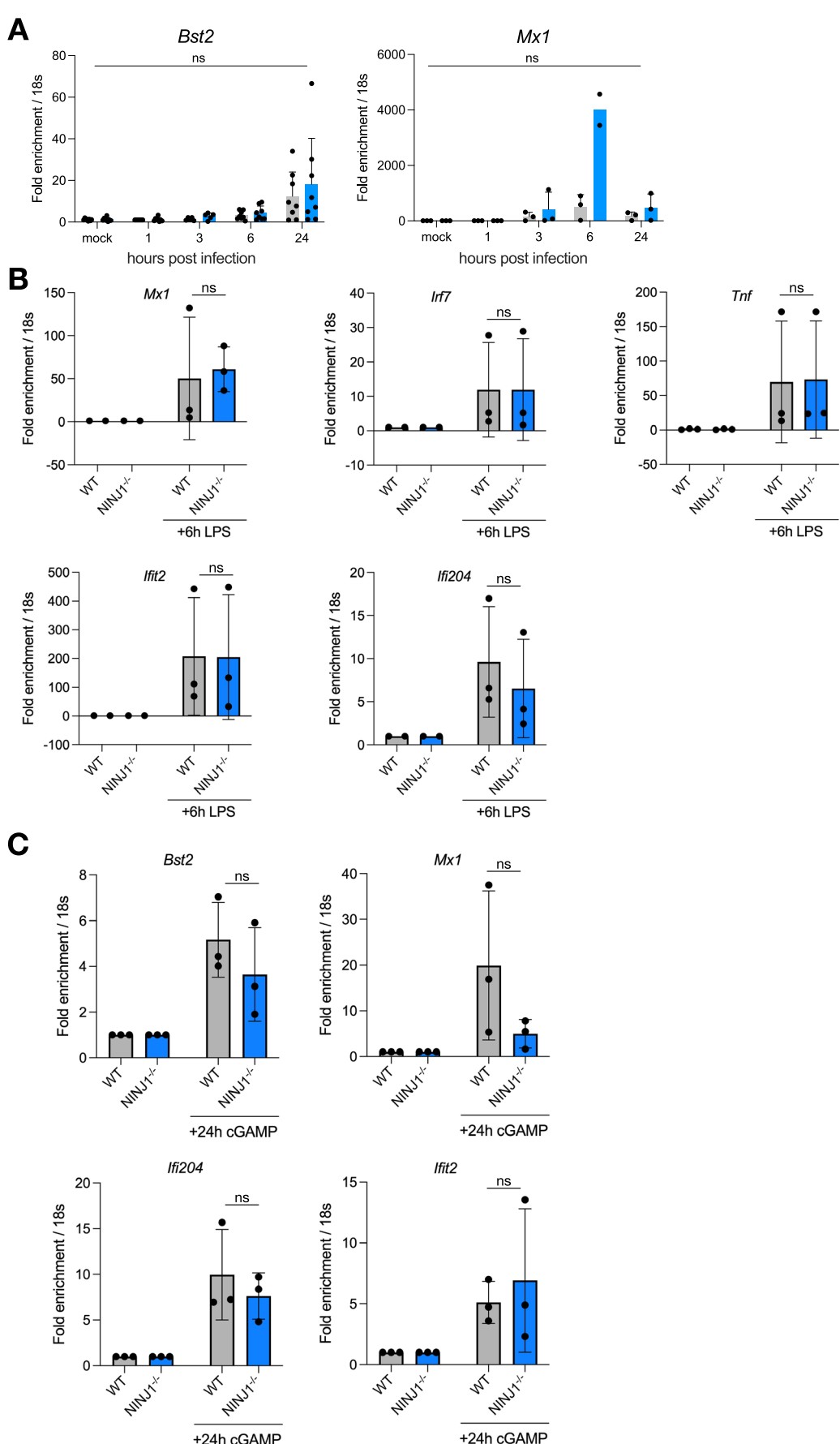

◀  **Figure EV3.  ISG transcript levels are higher in *Ninj1⁻/⁻ cells* upon HSV-1 infection.**

(**A**) WT and *Ninj1⁻/⁻* iBMDMs were infected for the indicated times with HSV-1 GFP, and RNA levels of the indicated transcripts were measured by RT-qPCR. (**B**) WT and *Ninj1⁻/⁻* iBMDMs ± stimulation with 10 μg/mL LPS for 6 h. RNA was harvested and subjected to qPCR for the indicated transcripts. (**C**) WT and *Ninj1⁻/⁻* iBMDMs ± stimulation with 5 μg/mL 2'3'cGAMP for 24 h. RNA was harvested and subjected to qPCR for the indicated transcripts. Data information: (**A–C**) mean ± SD is plotted with dots showing biological replicates, paired *t*-tests, ns not significant. (**A**) $n = 3$–8, (**B**, **C**) $n = 3$.

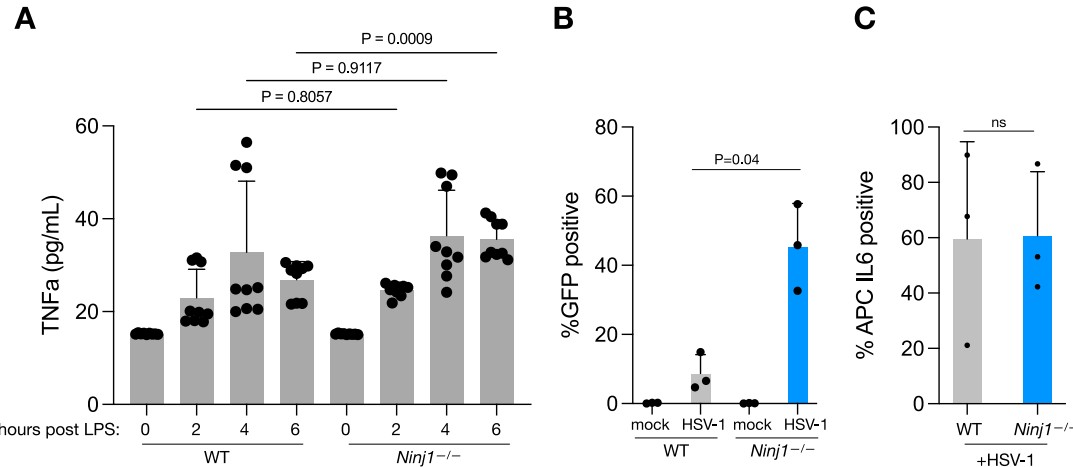

**Figure EV4. Loss of NINJ1 results in lower cytokine secretion.**

(**A**) WT and *Ninj1*⁻/⁻ iBMDMs were stimulated with 10 µg/mL LPS for the indicated periods of time. Supernatant were harvested and subjected to an ELISA for TNFα. Bars show the mean and SD. (**B, C**) WT and *Ninj1*⁻/⁻ iBMDMs were infected for 6 h and then stained for intracellular IL-6. Cells were analyzed on a flow cytometer for GFP levels (**B**) and APC IL-6 levels (**C**). Data information: (**A–C**) mean ± SD is plotted. A dots show three technical replicates from three biological replicates, Brown-Forsythe and Welch ANOVA. (**B, C**) dots show biological replicates, unpaired *t*-tests, ns not significant, *n* = 3.

