## [Peer Review File · EMBO Reports]

NINJ1 blocks HSV-1 entry into macrophages to impact viral replication and immunity

Ella Hartenian, Magalie Agustoni, and Petr Broz

Corresponding author(s): Petr Broz (petr.broz@unil.ch)

Review Timeline:

Transfer Date:	10th Feb 25
Editorial Decision:	11th Feb 25
Revision Received:	10th Jul 25
Editorial Decision:	18th Sep 25
Revision Received:	14th Oct 25
Editorial Decision:	15th Oct 25
Revision Received:	21st Oct 25
Accepted:	30th Oct 25

Editor: Achim Breiling / Martina Rembold

Transaction Report: This manuscript was transferred to EMBO reports following peer review at The EMBO Journal.

Dear Prof. Broz,

Thank you for transferring your manuscript to EMBO reports. I now went through the manuscript again, the referee reports from The EMBO Journal (attached again below) and your revision plan (provisional point-by-point response).

As discussed, judging from the revision plan, it seems that you will be able to adequately address the referee concerns during revision to allow publication at EMBO reports.

I thus invite you to revise your manuscript accordingly with the understanding that all referee concerns must be addressed in the revised manuscript and/or in a final detailed point-by-point response, as indicated in your letter.

Revised manuscripts should be submitted within three months of a request for revision. Please contact me to discuss the revision (also by video chat) if you have further questions or comments regarding the revision, or should you need additional time.

Acceptance of your manuscript will depend on a positive outcome of another round of review at EMBO reports, using the same referees.

- 1) a .docx formatted version of the final manuscript text (including legends for main figures, EV figures and tables), but without the figures included. Please make sure that changes are highlighted to be clearly visible. Figure legends should be compiled at the end of the manuscript text.
- 2) individual production quality figure files as .eps, .tif, .jpg (one file per figure), of main figures and EV figures. Please upload these as separate, individual files upon re-submission. Please make sure that all figure panels are called out separately and sequentially in the manuscript text

For more details please refer to our guide to authors:

See also our guide for figure preparation:

Moreover, please consult our guidelines for figure legend preparation:

- 4) a complete author checklist, which you can download from our author guidelines (<https://www.embopress.org/page/journal/14693178/authorguide>). Please insert page numbers in the checklist to indicate where the requested information can be found in the manuscript. The completed author checklist will also be part of the RPF.

5) that primary datasets produced in this study (e.g. RNA-seq, CHIP-seq and array data) are deposited in an appropriate public database. This is now mandatory (like the COI statement). If no primary datasets have been deposited in any database, please state this in this section (e.g. 'No primary datasets have been generated and deposited').

The accession numbers and database should be listed in a formal "Data Availability " section (placed after Materials & Methods) that follows the model below. Please note that the Data Availability Section is restricted to new primary data that are part of this study.

Data availability

8) Regarding data quantification and statistics, please make sure that the number "n" for how many independent experiments were performed, their nature (biological versus technical replicates), the bars and error bars (e.g. SEM, SD) and the test used to calculate p-values is indicated in the respective figure legends (also for potential EV figures and all those in the final Appendix). Please also check that all the p-values are explained in the legend, and that these fit to those shown in the figure. Please provide statistical testing where applicable. Please avoid the phrase 'independent experiment' but clearly state if these were biological or technical replicates. Please also indicate (e.g. with n.s.) if testing was performed, but the differences are not significant. In case n=2, please show the data as separate datapoints without error bars and statistics.

See also:

<http://www.embopress.org/page/journal/14693178/authorguide#statisticalanalysis>

If n<5, please show single datapoints for diagrams. Please add to each legend (main, EV figures, Appendix, where applicable) a 'Data Information' section explaining the statistics used or providing information regarding replicates and scales. See: <https://www.embopress.org/page/journal/14693178/authorguide#figureformat>

9) Please add scale bars of similar style and thickness to microscopic images, using clearly visible black or white bars (depending on the background). Please place these in the lower right corner of the images themselves. Please do not write on or near the bars in the image but define the size in the respective figure legend.

10) Please note our reference format:

11) We updated our journal's competing interests policy in January 2022 and request authors to consider both actual and perceived competing interests. Please review the policy <https://www.embopress.org/competing-interests> and add a statement declaring your competing interests. Please name that section 'Disclosure and Competing Interests Statement' and add it after the author contributions section.

12) Please order the sections like this using these names:

Title page - Abstract - Keywords - Introduction - Results - Discussion - Methods - Data availability section (DAS) - Acknowledgements (including funding information) - Disclosure and Competing Interests Statement - References - Figure legends - Expanded View Figure legends

13) Please provide the abstract written in present tense throughout.

14) Please make sure that all the funding information is also entered into the online submission system and is complete and similar to the one in the manuscript text file (in the Acknowledgements).

15) We now use CRediT to specify the contributions of each author in the journal submission system. CRediT replaces the author contribution section. Please use the free text box to provide more detailed descriptions. Thus, please do NOT provide your final manuscript text file with an author contributions section. See also guide to authors: <https://www.embopress.org/page/journal/14693178/authorguide#authorshipguidelines>

16) All materials and methods used need to be described in the main text using our 'Structured Methods' format, which is required for all research articles. According to this format, the Methods section should include a Reagents and Tools Table (listing key reagents, experimental models, software, and relevant equipment and including their sources and relevant identifiers), uploaded as separate file, followed by a Methods section in which we encourage the authors to describe their methods using a step-by-step protocol format with bullet points, to facilitate the adoption of the methodologies across labs. More information on how to adhere to this format as well as downloadable templates (.doc or .xls) for the Reagents and Tools Table can be found in our author guidelines (section 'Structured Methods'):

I look forward to seeing a revised version of your manuscript when it is ready. Please let me know if you have questions or comments regarding the revision.

Kind regards,

Achim

Referee #1:

The article 'NINJ1 is a restriction factor for HSV-1 in mouse macrophages' by Hartenian et al. describes a negative effect of NINJ1 on early stages of HSV-1 infection. The authors find that immortalized murine bone marrow-derived macrophages (iBMDMs) from NINJ1 mice are more susceptible to HSV-1 infection than wt iBMDMs, resulting in a larger fraction of infected cells, enhanced virus release, and a stronger activation of interferon-stimulated genes.

The manuscript is clearly written, and enhanced infection in NINJ1 knockouts is convincingly demonstrated with one immortalized cell line, although the available data does not fully rule out that this effect is unique to the described cell line (see below). A potential additional role of NINJ1 on top of its function in regulated cell death is certainly interesting. However, lacking clear mechanistic insights, I would recommend publishing this manuscript in a more specialized journal.

Major points:

1. A major concern of this study is that the observed effect on HSV-1 infection is only shown in one cell line derived from Ninj1^{-/-} BMDMs that underwent spontaneous immortalization. It is difficult to compare WT and Ninj1^{-/-} iBMDMs as the mutations or cellular alterations that allow continuous division may be different between two cell lines (and no additional clones/independently generated immortalized lines were compared). Importantly, the authors show that inducible expression of NINJ1 reduces HSV-1 infection and therefore rescues restriction. This suggests that NINJ1-independent differences in the two cell lines do not account for the observed differences. However, as virus-mediated gene expression is used as the sole readout, it cannot be ruled out that strong overexpression of any gene affects GFP expression. This experiment would therefore need controls in which another irrelevant gene is overexpressed and GFP levels are not affected (we have observed that overexpression of a random gene alters expression of virus-encoded reporters in our lab). Loss of NINJ1 did not affect infection in fibroblasts, ruling out any direct inhibitory effect of NINJ1 independent of the iBMDM context. It would therefore be more convincing that the role of NINJ1 is universal if the same phenotype was observed in WTE and KO primary BMDMs, independent KOs of NINJ1 derived from wt iBMDMs, or in cells in which NINJ1 was knocked down by RNAi (and in the best case other cell lines/models).

2. The applied bulk plate reader measurements of GFP signal are convenient, but do not allow a clear distinction between the kinetics of viral gene expression and the fraction of infected cells. Figure 1A and B clearly show by flow cytometry that the fraction of infected cells is higher in Ninj1^{-/-} iBMDMs, while the level of EGFP expression is not clearly different. If more of the Ninj1^{-/-} iBMDMs are infected, it is to be expected that the different phases of gene expression are more sensitively detected by Western Blot (and no proof of sped up kinetics). It is equally trivial that the infected culture in this case release more infectious progeny, that a larger fraction of cells exposes annexin V, and that a stronger ISG response is triggered. The interpretations therefore have to be specified, as the data does not support a reduction of 'all stages of the HSV-1 lifecycle'. I would further recommend using flow cytometry as a more definitive readout for infection for the most important experiments (which would also conveniently allow correlation with NINJ1 levels using antibody staining).

3. Regarding the mechanism of action, it remains unclear what NINJ1 does, and whether its role is conserved beyond the observed differences in one cell line. This would be required to make the manuscript relevant enough for EMBO J. The authors find that more infectious virions can be recovered after endocytosis, indicating that there is enhanced uptake, or reduced degradation of virions in endosomes. They report a trend of more viral capsids in the cytosol, although it remains unclear if the accumulated data from three independent experiments would reveal the same trends if all three experiments were evaluated individually (no statistics are applied and strictly speaking, the assay does not distinguish bound and internalized intact virions from released cores). Unless the authors show that HSV-1 is taken up by macropinocytosis, it is less telling that the fluid phase marker dextran is taken up similarly in wt and Ninj1^{-/-} iBMDMs. The easier comparison to evaluate similar and different endocytosis pathways would be to test if other viruses are also affected by the loss of NINJ1 (which would also point at a more general antiviral mechanism). It is clear that all these mechanistic details cannot be resolved in a revision, but at the current stage, the observed difference is merely a potentially interesting observation that could not be traced down to any potential molecular model.

Minor points:

1. Figure 2A:

The data in figure 2A suggests that Pycard^{-/-} iBMDMs cannot be infected with HSV-1, although the authors do not comment on this. This needs to be explained, as it may also be interpreted as a potential risk that the different iBMDM cell lines may not be equally susceptible to infection due to differences during immortalization (assuming ASC does not play a role in the viral life cycle).

2. Figure 2E:

Why was exclusively this experiment done with primary BMDMs? Please explain. Due to the weak signal of oligomers in the Nigericin sample, it cannot be ruled out that oligomers detection was just not sensitive enough. A more informative experiment would be to do the reconstitution experiments with NINJ1 mutants that cannot oligomerize/induce cell death (as described in Degen et al.) to evaluate which function of NINJ1 was required to restrict infection.

3. Figure 5A:

While the text states that 'We observed a trend of a stronger induction of ISGs in the Ninj1^{-/-} cells with the most pronounced difference in Viperin expression and smaller differences in Bst2 and IFI2712a expression (Fig5a)', the actual data indicates a somewhat stronger transcription of Viperin in the WT than in the Ninj1^{-/-} cells (unless I misinterpret the data, in which case better explanations may be required). This needs to be clarified. Figure 5B merely shows (without any further statistics) that ISG transcription after HSV-1 infection relies on MyD88 or TRIF, as well as (partially) on STING. I am sure this has been investigated elsewhere in much more detail and depth and is likely not worth a main figure. It is not really assessed whether the stronger infection and subsequent ISG induction in Ninj1^{-/-} iBMDMs relies on these adaptors.

4. Page 7:

'we infected sgluciferase expressing control cells and CRISPR-generated NINJ1 KO cells.' I assume that cells expressing Cas9 and sgRNAs for luciferase or Ninj1 were compared, but the sentence is not very intuitive to understand. Please correct.

Referee #2:

The authors present an interesting observation, which is accompanied with convincing - but not sufficient - characterization. Also, the authors should invest more in developing the story at the level of why HSV-1 infection in macrophages (which is not a natural host for HSV-1) is important. I find the work to have potential to become impactful if fully developed.

1. Fig 3. The data identifying HSV-1 entry as the step in the replication cycle targeted by NINJ1 are clear. However, the authors should provide more mechanistic information. E.g. does NINJ1 interact with HVME/TNFSF14 to prevent viral entry?
2. Fig 4. The authors should include human primary monocytes and macrophages in the characterization of NINJ1 expression across cells/tissues.
3. Fig 5. The identification of elevated ISG expression/PRR signaling upon HSV-1 infection in NINJ1-deficient macrophages should be explored broader. Most notably it should include testing for more IFB/ISGs and also NF- κ B-induced genes (e.g. IL-6 and TNFa). As a control the authors should show that the response through these PRRs per se is not altered, e.g. through stimulation with cGAMP.
4. Macrophages are not an important cellular source for HSV-1 replication. Therefore, the restriction of HSV-1 in this cell type is in itself unlikely to affect the outcome of infection very much. However, the finding that Ninj1-deficient macrophages exhibit elevated response through PRRs upon HSV-1 infection is interesting, and likely the finding with most impact. It is very much in line with the discussed in a recent review (PMID: 39675007). The present work would gain if discussed in the context of the thoughts in this review, which should be cited.
5. Following the point above, I would suggest that the authors change the title to "NINJ1 restricts HSV-1 in mouse macrophage and limits activation of pattern recognition receptors" (or something similar).
6. The work would of course gain substantial impact, if the authors could confirm the presently mouse-cell-in-vitro-based findings in human macrophages and/or Ninj1^{-/-} mice.

Referee #3:

Hartenian et al. report that ninjurin-1 (NINJ1)-deficient macrophages are more susceptible to herpes simplex virus (HSV) infection and that the cycle proceeds more rapidly and yields higher titers than wt controls due to more efficient entry but no measurable difference in cell death patterns. The authors should better reflect on the relevance of mice as well as the use of mouse macrophages to model HSV infection, a human virus that is naturally restricted to epithelial cells and neurons in the immunocompetent natural host. Mouse herpes encephalitis models have implicated brain macrophage-like cells as relevant hosts, but this has not been extended to humans. Skin resident phagocytic Langerhans cells may encounter HSV in humans. There are well-recognized species differences in the biology of HSV in rodent vs its natural human host suggesting that the pursuit of any role for NINJ1 in virus biology include some studies in human cells with mutants made by CRIPR mutagenesis in relevant cell types.

In the Introduction, the sentences "HSV-1 initially infects keratinocytes and fibroblasts, making its way to axons which then act as a conduit for HSV-1 to attain access to the brain." and "Once inside the brain, HSV-1 establishes latency in sensory neurons and undergoes rounds of reactivation." Should be accurately revised to indicate sensory neurons are part of the peripheral nervous system between the skin and the CNS separated by synapses, with brain (meninges in mice) only infected by viral strains that are employed to model herpes encephalitis. Infection of the brain is typically lethal in mice and results in death or permanent damage when seen in newborn and immunocompromised humans.

The term ninjurin-1 (NINJ1) should be written out and abbreviated in the title and upon first use in the text.

In the Results, the critical data shown in Figure 1 points immediately to a change in susceptibility in Ninj1^{-/-} compared to wt macrophages, particularly with the use of iBMDMs. BL/6 mice and macrophages, in particular, from BL/6 mice show long-ago resistance to HSV infection compared to other mouse strains (Lopez, 1975, Lopez, 1980, Sarmiento, 1988). In a supportive supplemental figure (preferably), the quality of the HSV virus stock should be carefully documented. Authors indicate that "Cell free viral particles were combined with cells subjected to douncing, ultracentrifuged at 15 000 x g for 90 minutes and the viral pellet was resuspended in DMEM with 10% FCS and 1g/100mL of BSA" but should mention whether they removed nuclei by low speed sedimentation after Dounce homogenization. If they did not, the presence of nuclear material will impact results. Authors should also further clarify how uniform an MOI calculated as 10 based on plaque assay using Vero cells will be on mouse macrophages. Thus, some assessment of the percentage of infected macrophages following inoculation should be included in the manuscript. Primary BMDMs and iBMDMs are likely to show differential and variable susceptibility to infection, so both should be evaluated. The authors need to convince the reader that differences between iBMDMs from different mutant mouse strains are not arbitrary, particularly given the unique behavior of the iBMDMs from Ninj1^{-/-} mice. Furthermore, uniform infection conditions (where 100% of cells are infected) are critical in the type of experiments being pursued here. First, it is crucial to increase the MOI and determine whether differences noted are qualitative or quantitative. In addition, it is critical to distinguish the impact of viral infection from the impact of virus particles on cells that do not initiate infection but nevertheless induce interferons and other cell autonomous responses. Both types of phenomenon may be biologically important, but must be distinguished. The line of experiments in the submitted manuscript leave a lot of questions as the authors seek to dissect the initial infection differences of wt and Ninj1^{-/-} BMDMs.

Figure 1 shows an evaluation of "viral proteins from different kinetic classes - ICP4 (immediate early), ICP8, (early) and gD"; however, it is important to include a true late protein (gC) in addition to the leaky late gD in such experiments given the difference in dependency on viral DNA replication..

As the authors point out, macrophages are not considered a critical host cell for HSV which infects epithelial cells and supportive fibroblasts in skin as well as sensory neurons during natural infection, so it would be important to investigate susceptibility of other cells from NINJ1-deficient mice, particularly because they are generally more susceptible than mouse macrophages to this human pathogen. Authors conceptualize that because NINJ1 is "a plasma membrane resident protein highly expressed in the monocyte lineage of both human and mouse", such cells will show the strongest phenotype; however, the depth of information on this protein and its role in lysis is still accumulating, warranting the comparison of different cell types.

The data in Figure 1H is not particularly convincing and should be accompanied by other strategies that seek to establish whether the deficiency in NINJ1 can be complemented. It is certainly possible that other changes have occurred that make the Ninj1^{-/-} cells more susceptible to HSV, a situation where introduction of NINJ1 would have a minimal impact.

Time courses are shown without reflecting a linear time scale (for example, in Figure 2C and 2D) and also using a combination of bars and data points (for example, Figure 2B and 5A), and these should be depicted appropriately.

One final overlooked data that must be included here is comparison of the susceptibility of wt and Ninj1^{-/-} mice following inoculation with fully wt and virulent HSV (not KOS-ICP47-GFP). There two very important reasons for including this experiment: Demonstrating a difference in susceptibility of mice with another HSV strain broadens impact and KOS (or KOS recombinant) is itself not virulent.

Once the authors have a clear picture that the differences in susceptibility are due to NINJ1, the manuscript will take more complete objective form.

Referee #4:

The authors studied the role of NINJ1 in mouse bone marrow derived macrophage (BMDMs). They found that Ninj1^{-/-} BMDMs are more susceptible to HSV-1 infection than WT cells. They attributed the higher HSV-1 infection rate of Ninj1^{-/-} macrophages to enhanced viral entry, but not to abnormal cell death following infection. They proposed that enhanced HSV-1 infection and replication in Ninj1^{-/-} macrophages resulted in higher ISG expression and inflammation. The mechanisms of how does Ninj1 control HSV-1 entry to these mouse macrophages remain unclear.

Overall, it is an interesting observation. My main concerns are:

- 1) The mechanisms of how does Ninj1 control HSV-1 entry to BMDMs remain unclear. The involvement of Nknj1 in cell death-dependent HSV-1 infection control is not fully excluded. Although they observed no clear difference in terms of cell lysis in Nknj1^{-/-} versus WT cells till 24 hpi. They also observed 2 logs higher HSV-1 levels in Nknj1^{-/-} cells 24 hpi, and increased Annexin V signal also at 24 hpi. All these together, doesn't it indicate an impairment of cell lysis in Nknj1^{-/-} BMDMs, despite enhanced HSV-1 replication and apoptosis?
- 2) The impact of Ninj1^{-/-} during HSV-1 infection was not studied in mouse models in vivo.

More specific comments:

- 1) Fig 2E, 'a low level of NINJ1 dimers was detectable, but this was comparable to untreated controls. Indeed. However, how could they exclude that the basal level NINJ1 dimerization/oligomerization could still play a role in mediating HSV-1 induced cell lysis?
- 2) Fig 3A, D, 3F, complementation of the phenotypes shown in these figures by WT NINJ1 is needed to confirm that lack of NINJ1 is the cause of the enhanced viral entry.
- 3) Fig 5, the data shown in this figure is overall weak. More independent repeats of experiment are needed, in particular given the huge variability of the data points shown for the same genotype(s). Not only mRNA levels of the ISGs, but also the protein production levels of IFNs and at least some inflammatory cytokines should be measured using cell supernatants from the same experiment.

Minor comments:

- 1) Introduction: HSV-1 infects more than 1 in 5 people worldwide. Please look up.
- 2) Also in Introduction: 'HSV-1 initially infects...and fibroblasts'. Do they mean this virus infects fibroblasts in vivo in human or mouse? How fibroblasts also assist the virus making its way to the brain? Please quote a reference.

Comments from editor:

No further understanding of the underlying mechanism would be required, all referees' comments regarding the solidity of the main conclusions of the manuscript, technical concerns, and points related to the experimental design, the used model systems, and data presentation should be addressed for consideration of the work at EMBO reports. In particular, major points 1 of referee #1 (asking for additional evidence that the new role of NINJ1 is universal; use of more/other cell lines as indicated by the referee), also consistent with major point 2 of referee #2), and major point 2 of referee #1 should be sufficiently addressed. No in vivo data would be required. If you are interested in this offer, Achim would be happy to discuss with you the specific requirements further; please feel free to contact him directly with a brief revision plan at a.breiling@emboreports.org. To transfer your manuscript to EMBO reports, please use the transfer link at the bottom of this message.

We thank the reviewers for the helpful comments and the editor for his offer to transfer this submission to EMBO Reports, which we are happy to accept. Below we detail our modifications to the manuscript in light of reviewer's comments.

Reviewers:

Referee #1:

Major points:

1. A major concern of this study is that the observed effect on HSV-1 infection is only shown in one cell line derived from Ninj1^{-/-} BMDMs that underwent spontaneous immortalization. It is difficult to compare WT and Ninj1^{-/-} iBMDMs as the mutations or cellular alterations that allow continuous division may be different between two cell lines (and no additional clones/independently generated immortalized lines were compared). Importantly, the authors show that inducible expression of NINJ1 reduces HSV-1 infection and therefore rescues restriction. This suggests that NINJ1-independent differences in the two cell lines do not account for the observed differences. However, as virus-mediated gene expression is used as the sole readout, it cannot be ruled out that strong overexpression of any gene affects GFP expression. This experiment would therefore need controls in which another irrelevant gene is overexpressed and GFP levels are not affected (we have observed that overexpression of a random gene alters expression of virus-encoded reporters in our lab). Loss of NINJ1 did not affect infection in fibroblasts, ruling out any direct inhibitory effect of NINJ1 independent of the iBMDM context. It would therefore be more convincing that the role of NINJ1 is universal if the same phenotype was observed in WTE and KO primary BMDMs, independent KOs of NINJ1 derived from wt iBMDMs, or in cells in which NINJ1 was knocked down by RNAi (and in the best case other cell lines/models).

We thank the reviewer for their comments and for raising this issue.

We would like to emphasize that, in addition to using immortalized BMDMs, we also included data acquired in primary WT and NINJ1^{-/-} macrophages in our initial manuscript submission (see Fig. 1C). Importantly, we observed the same effect of NINJ1 deficiency, namely elevated viral replication, in both primary and immortalized cells. Therefore, the observed phenotype cannot be attributed to the process of immortalization.

To further strengthen this point, we have now included a panel showing a quantification of GFP by flow cytometry in these primary WT and NINJ1^{-/-} macrophages. This is now Figure 1D.

The reviewer is also concerned that the complementation in Figure 1H could be due to a more general effect of overexpressing a protein, instead of the complementation of the NINJ1-deficiency by exogenous expression of NINJ1.

This is a valid concern, and we have addressed it by expressing blasticidin resistance cassette lentivirally into the iNINJ1 cell line and comparing infection with and without expression of this transgene. We did not observe a change in replication kinetics, thus confirming that expression of an unrelated protein does not reduce HSV-1 replication. This is now Extended View Figure 1B.

2. The applied bulk plate reader measurements of GFP signal are convenient, but do not allow a clear distinction between the kinetics of viral gene expression and the fraction of infected cells. Figure 1A and B clearly show by flow cytometry that the fraction of infected cells is higher in Ninj1^{-/-} iBMDMs, while the level of EGFP expression is not clearly different. If more of the Ninj1^{-/-} iBMDMs are infected, it is to be expected that the different phases of gene expression are more sensitively detected by Western Blot (and no proof of sped up kinetics). It is equally trivial that the infected culture in this case release more infectious progeny, that a larger fraction of cells exposes annexin V, and that a stronger ISG response is triggered. The interpretations therefore have to be specified, as the data does not support a reduction of 'all stages of the HSV-1 lifecycle'. I would further recommend using flow cytometry as a more definitive readout for infection for the most important experiments (which would also conveniently allow correlation with NINJ1 levels using antibody staining).

We thank the reviewer for pointing this out and have changed the text accordingly. We do note that by MFI the level of EGFP expression is increased (Figure 1B) which we now quantify and include as Extended View Figure 1A.

3. Regarding the mechanism of action, it remains unclear what NINJ1 does, and whether its role is conserved beyond the observed differences in one cell line. This would be required to make the manuscript relevant enough for EMBO J. The authors find that more infectious virions can be recovered after endocytosis, indicating that there is enhanced uptake, or reduced degradation of virions in endosomes. They report a trend of more viral capsids in the cytosol, although it remains unclear if the accumulated data from three independent experiments would reveal the same trends if all three experiments were evaluated individually (no statistics are applied and strictly speaking, the assay does not distinguish bound and internalized intact virions from released cores).

Unless the authors show that HSV-1 is taken up by macropinocytosis, it is less telling that the fluid phase marker dextran is taken up similarly in wt and Ninj1^{-/-} iBMDMs. The easier comparison to evaluate similar and different endocytosis pathways would be to test if other viruses are also affected by the loss of NINJ1 (which would also point at a more general antiviral mechanism). It is clear that all these mechanistic details cannot be resolved in a revision, but at the current stage, the observed difference is merely a potentially interesting observation that could not be traced down to any potential molecular model.

We thank the reviewer for their comments. We have now included data with VACV, another enveloped virus that uses endocytosis and micropinocytosis to enter cells, and we also observed higher levels of intracellular virions after a 10 min infection in cells lacking NINJ1 compared to WT cells. This suggests that NINJ1 affect entry of different viruses, not only HSV-1. This is now Figure 3F.

Minor points:

1. Figure 2A:

The data in figure 2A suggests that Pycard^{-/-} iBMDMs cannot be infected with HSV-1, although the authors do not comment on this. This needs to be explained, as it may also be interpreted as a potential risk that the different iBMDM cell lines may not be equally susceptible to infection due to differences during immortalization (assuming ASC does not play a role in the viral life cycle).

We thank the reviewer for pointing this out. Infection of Pycard^{-/-} iBMDMs was consistently low in our experiments but not always 0. We have replaced the previous figure panel with another more representative experiment (now Fig 2A,B).

2. Figure 2E:

Why was exclusively this experiment done with primary BMDMs? Please explain. Due to the weak signal of oligomers in the Nigericin sample, it cannot be ruled out that oligomers detection was just not sensitive enough. A more informative experiment would be to do the reconstitution experiments with NINJ1 mutants that cannot oligomerize/induce cell death (as described in Degen et al.) to evaluate which function of NINJ1 was required to restrict infection.

Oligomerization of NINJ1 is an all or nothing readout. We either see oligomeric NINJ1 or not, giving us confidence that the weak signal is not masking a true oligomerization. We have also included additional replicates of this experiment in the source data, allowing readers to evaluate all of the data we collected.

The reviewer also suggest we do the reconstitution experiments with NINJ1 mutants as we did in Degen et al. We attempted to perform this experiment but were not successful. The main reason was that these reconstitutions in primary cells relies on an mCherry-positive MSCV vector delivering mutant NINJ1, which has low infectivity, resulting in only 30–40% of macrophages cells actually expressing the protein. Subsequent infection with HSV-1 GFP yielded a very small fraction of double-positive cells, rendering the results

uninterpretable. Repeating this experiment would be lengthy and challenging and our preliminary results did not suggest we would be successful.

3. Figure 5A:

While the text states that 'We observed a trend of a stronger induction of ISGs in the Ninj1^{-/-} cells with the most pronounced difference in Viperin expression and smaller differences in Bst2 and IFI2712a expression (Fig5a)', the actual data indicates a somewhat stronger transcription of Viperin in the WT than in the Ninj1^{-/-} cells (unless I misinterpret the data, in which case better explanations may be required). This needs to be clarified. Figure 5B merely shows (without any further statistics) that ISG transcription after HSV-1 infection relies on MyD88 or TRIF, as well as (partially) on STING. I am sure this has been investigated elsewhere in much more detail and depth and is likely not worth a main figure. It is not really assessed whether the stronger infection and subsequent ISG induction in Ninj1^{-/-} iBMDMs relies on these adaptors.

We thank the reviewer for this comment. The legend for this figure was switched between WT and NINJ1 and it is indeed the RNA level of ISGs in the NINJ1^{-/-} condition that are higher and this has now been fixed.

Indeed, the role of TLR and NFKb signaling in response to HSV-1 infection has been examined in other cell types, but never in macrophages. Thus, we are providing the first investigation of the sensing pathways employed by macrophages.

4. Page 7:

'we infected sgLuciferase expressing control cells and CRISPR-generated NINJ1 KO cells.' I assume that cells expressing Cas9 and sgRNAs for Luciferase or Ninj1 were compared, but the sentence is not very intuitive to understand. Please correct.

This has been clarified with text changes.

Referee #2:

The authors present an interesting observation, which is accompanied with convincing - but not sufficient - characterization. Also, the authors should invest more in developing the story at the level of why HSV-1 infection in macrophages (which is not a natural host for HSV-1) is important. I find the work to have potential to become impactful if fully developed.

1. Fig 3. The data identifying HSV-1 entry as the step in the replication cycle targeted by NINJ1 are clear. However, the authors should provide more mechanistic information. E.g. does NINJ1 interact with HVME/TNFSF14 to prevent viral entry?

We thank the reviewer for this suggestion. Since we have only limited quantities of anti-NINJ1 antibody, we could not do immunoprecipitation/mass spectrometry for the endogenous protein. We performed however a GFP-trap/mass spectrometry on

overexpressed NINJ1-GFP to identify interaction partners of NINJ1 in HeLa cells. However, we did not detect either HVEM or Nectin 1 to reliable levels in this cell line, thus we would prefer not to include these data in the study.

2. Fig 4. The authors should include human primary monocytes and macrophages in the characterization of NINJ1 expression across cells/tissues.

We thank the reviewer for this comment, however we believe we address this comment already as in Figure 4 we include human monocyte derived macrophages (hMDM, first lane) and a human monocyte cell line (U937). We have now further bolstered this figure by measuring NINJ1 protein expression across 12 different mouse tissues (Figure 4C, EV 2).

3. Fig 5. The identification of elevated ISG expression/PRR signaling upon HSV-1 infection in NINJ1-deficient macrophages should be explored broader. Most notably it should include testing for more IFB/ISGs and also NF- κ B-induced genes (e.g. IL-6 and TNFa). As a control the authors should show that the response through these PRRs per se is not altered, e.g. through stimulation with cGAMP.

We thank the reviewer for this suggestion. We have included additional transcripts in Figure 5A and Extended Figure 3, including TNFa and IFNB. We also treated WT and *Ninj1*^{-/-} cells with LPS and cGAMP and compared induction of ISGs to show that there are no intrinsic differences. These are now included as EV 3B-C.

4. Macrophages are not an important cellular source for HSV-1 replication. Therefore, the restriction of HSV-1 in this cell type is in itself unlikely to affect the outcome of infection very much. However, the finding that *Ninj1*-deficient macrophages exhibit elevated response through PRRs upon HSV-1 infection is interesting, and likely the finding with most impact. It is very much in line with the discussed in a recent review (PMID: 39675007). The present work would gain if discussed in the context of the thoughts in this review, which should be cited.

This is a nice suggestion and a thought-provoking review, and we have incorporated a discussion of it in the manuscript, including a citation.

5. Following the point above, I would suggest that the authors change the title to "NINJ1 restricts HSV-1 in mouse macrophage and limits activation of pattern recognition receptors" (or something similar).

Thanks for this suggestion, we have changed the title to reflect the impact of NINJ1 on the immune response.

6. The work would of course gain substantial impact, if the authors could confirm the presently mouse-cell-in-vitro-based findings in human macrophages and/or *Ninj1*^{-/-} mice.

We agree with the reviewer, however the logistics of performing mouse experiments with HSV-1 using an eye scarification model are challenging in Switzerland where it would take us more than a year to request and obtain an infection license. In our hands neither U937 nor THP1 human monocyte lines express significant levels of NINJ1 thus the relevance of knocking out Ninj1 in this context is questionable.

Referee #3:

Hartenian et al. report that ninjurin-1 (NINJ1)-deficient macrophages are more susceptible to herpes simplex virus (HSV) infection and that the cycle proceeds more rapidly and yields higher titers than wt controls due to more efficient entry but no measurable difference in cell death patterns. The authors should better reflect on the relevance of mice as well as the use of mouse macrophages to model HSV infection, a human virus that is naturally restricted to epithelial cells and neurons in the immunocompetent natural host. Mouse herpes encephalitis models have implicated brain macrophage-like cells as relevant hosts, but this has not been extended to humans. Skin resident phagocytic Langerhans cells may encounter HSV in humans. There are well-recognized species differences in the biology of HSV in rodent vs its natural human host suggesting that the pursuit of any role for NINJ1 in virus biology include some studies in human cells with mutants made by CRIPR mutagenesis in relevant cell types.

We thank the reviewer for this question. Similar to our response to the previous reviewer, neither U937 nor THP1 human monocytes lines express significant levels of NINJ1 thus the relevance of knocking out Ninj1 in this context is questionable. We have observed NINJ1 expression in human primary tissue (ie hMDMs), but obtaining and manipulating these are beyond the scope of this study.

In the Introduction, the sentences "HSV-1 initially infects keratinocytes and fibroblasts, making its way to axons which then act as a conduit for HSV-1 to attain access to the brain." and "Once inside the brain, HSV-1 establishes latency in sensory neurons and undergoes rounds of reactivation." Should be accurately revised to indicate sensory neurons are part of the peripheral nervous system between the skin and the CNS separated by synapses, with brain (meninges in mice) only infected by viral strains that are employed to model herpes encephalitis. Infection of the brain is typically lethal in mice and results in death or permanent damage when seen in newborn and immunocompromised humans.

Thank you for this clarification, we have changed the text accordingly.

The term ninjurin-1 (NINJ1) should be written out and abbreviated in the title and upon first use in the text.

This has been modified in the title and upon the first use in the text.

In the Results, the critical data shown in Figure 1 points immediately to a change in susceptibility in Ninj1^{-/-} compared to wt macrophages, particularly with the use of

iBMDMs. BL/6 mice and macrophages, in particular, from BL/6 mice show long-ago resistance to HSV infection compared to other mouse strains (Lopez, 1975, Lopez, 1980, Sarmiento, 1988). In a supportive supplemental figure (preferably), the quality of the HSV virus stock should be carefully documented. Authors indicate that "Cell free viral particles were combined with cells subjected to douncing, ultracentrifuged at 15 000 x g for 90 minutes and the viral pellet was resuspended in DMEM with 10% FCS and 1g/100mL of BSA" but should mention whether they removed nuclei by low speed sedimentation after Dounce homogenization.

We thank the reviewer for the comments and have clarified in the methods that we did perform low speed centrifugation.

If they did not, the presence of nuclear material will impact results. Authors should also further clarify how uniform an MOI calculated as 10 based on plaque assay using Vero cells will be on mouse macrophages. Thus, some assessment of the percentage of infected macrophages following inoculation should be included in the manuscript. Primary BMDMs and iBMDMs are likely to show differential and variable susceptibility to infection, so both should be evaluated. The authors need to convince the reader that differences between iBMDMs from different mutant mouse strains are not arbitrary, particularly given the unique behavior of the iBMDMs from *Ninj1*^{-/-} mice. Furthermore, uniform infection conditions (where 100% of cells are infected) are critical in the type of experiments being pursued here. First, it is crucial to increase the MOI and determine whether differences noted are qualitative or quantitative. In addition, it is critical to distinguish the impact of viral infection from the impact of virus particles on cells that do not initiate infection but nevertheless induce interferons and other cell autonomous responses. Both types of phenomenon may be biologically important, but must be distinguished. The line of experiments in the submitted manuscript leave a lot of questions as the authors seek to dissect the initial infection differences of wt and *Ninj1*^{-/-} BMDMs.

We thank the reviewer for these comments. In our experience, it is not possible to infect 100% of iBMDMs or BMDMs and we are already functioning at some of the highest levels of infection we can obtain. We have now quantified infection of primary macrophages by flow cytometry and include this as Figure 1D.

Figure 1 shows an evaluation of "viral proteins from different kinetic classes - ICP4 (immediate early), ICP8, (early) and gD"; however, it is important to include a true late protein (gC) in addition to the leaky late gD in such experiments given the difference in dependency on viral DNA replication..

We thank the reviewer for these comments. We think there must be a misunderstanding as in Figure 3A we see no dependency on viral DNA replication by using the viral DNA replication inhibitor, PAA. Nevertheless we have attempted to detect gC with abcam antibody ab6509, however, it did not give any signal. Thus, we have clarified that gD is a leaky late gene in the text.

As the authors point out, macrophages are not considered a critical host cell for HSV which infects epithelial cells and supportive fibroblasts in skin as well as sensory neurons during natural infection, so it would be important to investigate susceptibility of other cells from NINJ1-deficient mice, particularly because they are generally more susceptible than mouse macrophages to this human pathogen. Authors conceptualize that because NINJ1 is "a plasma membrane resident protein highly expressed in the monocyte lineage of both human and mouse", such cells will show the strongest phenotype; however, the depth of information on this protein and its role in lysis is still accumulating, warranting the comparison of different cell types.

We have already tried infection of WT and Ninj1 ko MEFs, and isolation and establishment of a primary cell culture from other organs would take significant optimization that is beyond the scope of this paper.

Thus, given that little is known about the cell types in which NINJ1 is expressed, we have compared tissue expression of NINJ1 across mouse organs between WT and Ninj1^{-/-} animals to clarify where NINJ1 could influence infection. This is now Figure 4C and EV 2.

The data in Figure 1H is not particularly convincing and should be accompanied by other strategies that seek to establish whether the deficiency in NINJ1 can be complemented. It is certainly possible that other changes have occurred that make the Ninj1^{-/-} cells more susceptible to HSV, a situation where introduction of NINJ1 would have a minimal impact.

We thank the reviewer for their comments and have bolstered the complementation by the control experiment that Reviewer 1 proposed, showing that transgene expression alone is not sufficient to alter GFP levels. This is now EV 1B.

Time courses are shown without reflecting a linear time scale (for example, in Figure 2C and 2D) and also using a combination of bars and data points (for example, Figure 2B and 5A), and these should be depicted appropriately.

Thank you for this comment, we have changed the figures accordingly.

One final overlooked data that must be included here is comparison of the susceptibility of wt and Ninj1^{-/-} mice following inoculation with fully wt and virulent HSV (not KOS-ICP47-GFP). There two very important reasons for including this experiment: Demonstrating a difference in susceptibility of mice with another HSV strain broadens impact and KOS (or KOS recombinant) is itself not virulent.

We agree with the reviewer, however the logistics of performing mouse experiments with HSV-1 using an eye scarification model are challenging in Switzerland where it would take us more than a year to request and obtain an infection license.

Once the authors have a clear picture that the differences in susceptibility are due to NINJ1, the manuscript will take more complete objective form.

Referee #4:

The authors studied the role of NINJ1 in mouse bone marrow derived macrophage (BMDMs). They found that *Ninj1*^{-/-} BMDMs are more susceptible to HSV-1 infection than WT cells. They attributed the higher HSV-1 infection rate of *Ninj1*^{-/-} macrophages to enhanced viral entry, but not to abnormal cell death following infection. They proposed that enhanced HSV-1 infection and replication in *Ninj1*^{-/-} macrophages resulted in higher ISG expression and inflammation. The mechanisms of how does *Ninj1* control HSV-1 entry to these mouse macrophages remain unclear.

Overall, it is an interesting observation. My main concerns are:

1) The mechanisms of how does *Ninj1* control HSV-1 entry to BMDMs remain unclear. The involvement of *Nknj1* in cell death-dependent HSV-1 infection control is not fully excluded. Although they observed no clear difference in terms of cell lysis in *Nknj1*^{-/-} versus WT cells till 24 hpi. They also observed 2 logs higher HSV-1 levels in *Nknj1*^{-/-} cells 24 hpi, and increased Annexin V signal also at 24 hpi. All these together, doesn't it indicate an impairment of cell lysis in *Nknj1*^{-/-} BMDMs, despite enhanced HSV-1 replication and apoptosis?

We thank the reviewer for this comment. We do not observe any oligomerization of NINJ1 and thus can exclude a cell-death dependent role of NINJ1 in restricting HSV-1 (oligomerization is absolutely needed for NINJ1-driven cell lysis). It is correct that we observe higher AnnV levels in NINJ1-deficient cells, but this is most likely a consequence of the enhanced replication of HSV-1 in absence of NINJ1, and not a consequence of a lysis defect in NINJ1-deficient cells.

2) The impact of *Ninj1*^{-/-} during HSV-1 infection was not studied in mouse models in vivo.

We agree with the reviewer that this represents an interesting and exciting extension of this study, however the logistics of performing mouse experiments with HSV-1 using an eye scarification model are challenging in Switzerland where it would take us more than a year to request and obtain an infection license.

More specific comments:

1) Fig 2E, 'a low level of NINJ1 dimers was detectable, but this was comparable to untreated controls. Indeed. However, how could they exclude that the basal level NINJ1 dimerization/oligomerization could still play a role in mediating HSV-1 induced cell lysis?

We think this is an interesting suggestion and indeed expect that NINJ1 dimers could play a role at resting state, potentially in altering the fluidity of the membrane. We have further discussed this in the text.

2) Fig 3A, D, 3F, complementation of the phenotypes shown in these figures by WT NINJ1 is needed to confirm that lack of NINJ1 is the cause of the enhanced viral entry.

We thank the reviewer for this suggestion. We have addressed the complementation by expressing a blasticidin resistance cassette into the iNINJ1 cell line using lentiviral and comparing infection with and without expression of this transgene. We did not observe a change in replication kinetics, thus confirming that expression of an unrelated protein does not reduce HSV-1 replication. This is now Extended View 1B.

3) Fig 5, the data shown in this figure is overall weak. More independent repeats of experiment are needed, in particular given the huge variability of the data points shown for the same genotype(s). Not only mRNA levels of the ISGs, but also the protein production levels of IFNs and at least some inflammatory cytokines should be measured using cell supernatants from the same experiment.

We thank the reviewer for this suggestion. As we indicated to a similar point made by reviewer 2, we have performed a multiplex analysis to look at protein levels of a panel of cytokines in response to infection. This is now Figure 5C.

Minor

comments:

1) Introduction: HSV-1 infects more than 1 in 5 people worldwide. Please look up.

Thank you for pointing out this typo, we have fixed the text accordingly.

2) Also in Introduction: 'HSV-1 initially infects...and fibroblasts'. Do they mean this virus infects fibroblasts in vivo in human or mouse? How fibroblasts also assist the virus making its way to the brain? Please quote a reference.

Thanks for this correction, we have changed the text accordingly.

Dear Prof. Broz

Thank you for the submission of your revised manuscript to EMBO reports. I apologize for the delay in handling it, which was caused by general delays in summer time in the review process and an understaffing of our editorial office early September. Since my colleague Achim is currently out of office, I have temporarily taken over the handling of your manuscript. We have meanwhile received the full set of referee reports that is copied below.

As you will see, the referees find that the revised manuscript has been considerably strengthened and that all concerns were overall addressed adequately, except for providing more mechanism. Since a detailed mechanistic understanding is not required for publication at EMBO reports, we will proceed with publication. Please address the concerns from referee #3 regarding the validity of the mouse model and relevance to human HSV pathogenesis in the manuscript, clearly discussing these limitations of your study. The remaining concerns from referee #4 should also be addressed.

Please provide a point-by-point response to the referee concerns and also the editorial points (see below).

From the editorial side, there are also a few things that we need before we can proceed with the official acceptance of your study.

- You manuscript will be published in our Reports section. Please combine the Results and Discussion sections and keep an eye on the character count, which should be approx. 27,000 characters incl. spaces but excluding methods and references.

- Please add the corresponding author's email address on the title page.

- Please add a callout for Fig. 4F in the text.

- Please upload the source data as one folder (zipped) per figure.

- Please add the reference to the source data images at BioStudies in the Data Availability section and please add a URL that resolves directly to the dataset.

- Materials and Methods should be Methods

- Please correct the labels on the EV figures to Expanded View Figure #.

- Also in the Expanded View Figure Legends the names need to be corrected to Figure EV# instead of Expanded View 1.

- Please define the error bars in the legends of figures 2D, E; 3G.

- Please specify the exact p values in the legend of figure 1A.

- Please indicate the statistical test used for data analysis in the legends of figures 1D, 3G, I"

- Reagent table: please remove the Instructions paragraph.

- You cite two preprints, Ding X et al 2024 and Hartenian and Bernard 2025.

The correct citation format for these is

(preprint: Ding X et al, 2025) in the manuscript text

Addition of [PREPRINT] at the end of the reference in the reference list.

For Hartenian and Bernard, 2025 I noted that it is not part of the reference list, it is only an in-text citation.

- Titles should not contain punctuation. What about: "NINJ1 blocks HSV-1 entry into macrophages to impact viral replication and immunity"

- Please write the abstract in present tense.

- Finally, EMBO Reports papers are accompanied online by

A) a short (1-2 sentences) summary of the findings and their significance,

B) 2-3 bullet points highlighting key results and

C) a schematic summary figure that provides a sketch of the major findings (not a data image).

Please provide the summary figure as a separate file in PNG or JPG format at a size of 550x300-600 pixels (width x height).

Please note that the size is rather small and that text needs to be readable at the final size. Please send us this information along with the revised manuscript.

With kind regards,

=====

Referee #1:

I find that the reviewers have addressed the points in a satisfactory manner, and that the conclusions are now fully supported by the data.

Referee #2:

The authors have adequately addressed the major comments from the reviewers as requested by the editor. They have replied to each of my specific minor comments as well.

Referee #3:

Hartenian et al. have revised their initial manuscript showing that ninjurin-1 (NINJ1), a macrophage membrane protein known to function in cell adhesion and cell death, restricts herpes simplex virus (HSV) at the entry stages of infection, apparently reducing the number of viral particles capable of entering murine macrophages. In the revised manuscript, they show that vaccinia virus is similarly restricted. The manuscript does not bring a mechanistic understanding of this restriction and will be of interest mostly to specialists.

In the response to initial review, authors have undertaken only modest revision, mostly to correct initial experimental shortcomings, but have not fundamentally increased mechanistic understanding or brought forward the relevance of their observations to HSV pathogenesis. The murine model of HSV is controversial but nevertheless has implicated both resident microglia (Reinhart et al, 2016 and 2021) as well as monocyte-derived macrophages (Katzilieris-Petras et al., 2022) in the brain in control of murine HSV encephalitis (significant work from the Paludan group that deserves better referencing and integration here). In addition, mouse monocyte-derived cell types are recruited to skin and become viral antigen positive following HSV infection (Eidsmo et al., 2009). Human monocyte-derived macrophages have also been a target of HSV study (Melchjorsen et al., 2006) and possibly control HSV along with virus-specific T cells in skin. Unfortunately, the relevance of murine bone marrow-derived macrophages to any setting of HSV infection or disease has not been established and the known complexities of HSV restriction in various mouse cells require further mechanistic insight and relevance in human cells.

Referee #4:

Hartenian et al. submitted a revised version to EMBO Reports, which addressed some of the reviewer's concerns and still describes an interesting finding that could not yet be mechanistically dissected. Overall, the manuscript meets the scope of EMBO Reports and can be published after minor revisions as pointed out below (my responses to the answers in the rebuttal letter are indicated with a '>').

Referee #1: Major points: 1. A major concern of this study is that the observed effect on HSV-1 infection is only shown in one cell line derived from Ninj1^{-/-} BMDMs that underwent spontaneous immortalization. It is difficult to compare WT and Ninj1^{-/-} iBMDMs as the mutations or cellular alterations that allow continuous division may be different between two cell lines (and no additional clones/independently generated immortalized lines were compared). Importantly, the authors show that inducible expression of NINJ1 reduces HSV-1 infection and therefore rescues restriction. This suggests that NINJ1-independent differences in the two cell lines do not account for the observed differences. However, as virus-mediated gene expression is used as the sole readout, it cannot be ruled out that strong overexpression of any gene affects GFP expression. This experiment would therefore need controls in which another irrelevant gene is overexpressed and GFP levels are not affected (we have observed that overexpression of a random gene alters expression of virus-encoded reporters in our lab). Loss of NINJ1 did not

affect infection in fibroblasts, ruling out any direct inhibitory effect of NINJ1 independent of the iBMDM context. It would therefore be more convincing that the role of NINJ1 is universal if the same phenotype was observed in WTE and KO primary BMDMs, independent KOs of NINJ1 derived from wt iBMDMs, or in cells in which NINJ1 was knocked down by RNAi (and in the best case other cell lines/models).

We thank the reviewer for their comments and for raising this issue.

We would like to emphasize that, in addition to using immortalized BMDMs, we also included data acquired in primary WT and NINJ1^{-/-} macrophages in our initial manuscript submission (see Fig. 1C). Importantly, we observed the same effect of NINJ1 deficiency, namely elevated viral replication, in both primary and immortalized cells. Therefore, the observed phenotype cannot be attributed to the process of immortalization.

To further strengthen this point, we have now included a panel showing a quantification of GFP by flow cytometry in these primary WT and NINJ1^{-/-} macrophages. This is now Figure 1D.

>Fig. 1C did not contain quantitative data and it is very subjective to judge which zoomed in area of the culture is representative. The new figure 1D answers the reviewer's concerns and shows that the described effect is also observed in primary cells that are not immortalized. The 'representative' microscopy images in 1C do not seem to show the same percentages of infected cells as the flow cytometry data in 1D. Please comment if any experimental conditions were different.

The reviewer is also concerned that the complementation in Figure 1H could be due to a more general effect of overexpressing a protein, instead of the complementation of the NINJ1-deficiency by exogenous expression of NINJ1.

This is a valid concern, and we have addressed it by expressing blasticidin resistance cassette lentivirally into the iNINJ1 cell line and comparing infection with and without expression of this transgene. We did not observe a change in replication kinetics, thus confirming that expression of an unrelated protein does not reduce HSV-1 replication. This is now Extended View Figure 1B.

>This reviewer point was addressed, although it would have been more accurate to use a lentiviral construct with the same promoter. If I understand it correctly, pLVX mNinj1 uses a dox-inducible promoter, while pLJM1 carries a CMV promoter. Please update the plasmid information in the reagent table to include both vectors.

2. The applied bulk plate reader measurements of GFP signal are convenient, but do not allow a clear distinction between the kinetics of viral gene expression and the fraction of infected cells. Figure 1A and B clearly show by flow cytometry that the fraction of infected cells is higher in Ninj1^{-/-} iBMDMs, while the level of EGFP expression is not clearly different. If more of the Ninj1^{-/-} iBMDMs are infected, it is to be expected that the different phases of gene expression are more sensitively detected by Western Blot (and no proof of sped up kinetics). It is equally trivial that the infected culture in this case release more infectious progeny, that a larger fraction of cells exposes annexin V, and that a stronger ISG response is triggered. The interpretations therefore have to be specified, as the data does not support a reduction of 'all stages of the HSV-1 lifecycle'. I would further recommend using flow cytometry as a more definitive readout for infection for the most important experiments (which would also conveniently allow correlation with NINJ1 levels using antibody staining).

We thank the reviewer for pointing this out and have changed the text accordingly. We do note that by MFI the level of EGFP expression is increased (Figure 1B) which we now quantify and include as Extended View Figure 1A.

>Fig. EV1A shows a substantial change in one of the four quantified experiments, and I guess this can very much depend on the gates chosen for the quantification. In the shown representative curve, this is not too obvious. I would suggest rephrasing the sentence and toning down the conclusions on the MFI.

3. Regarding the mechanism of action, it remains unclear what NINJ1 does, and whether its role is conserved beyond the observed differences in one cell line. This would be required to make the manuscript relevant enough for EMBO J. The authors find that more infectious virions can be recovered after endocytosis, indicating that there is enhanced uptake, or reduced degradation of virions in endosomes. They report a trend of more viral capsids in the cytosol, although it remains unclear if the accumulated data from three independent experiments would reveal the same trends if all three experiments were evaluated individually (no statistics are applied and strictly speaking, the assay does not distinguish bound and internalized intact virions from released cores).

Unless the authors show that HSV-1 is taken up by macropinocytosis, it is less telling that the fluid phase marker dextran is taken up similarly in wt and Ninj1^{-/-} iBMDMs. The easier comparison to evaluate similar and different endocytosis pathways would be to test if other viruses are also affected by the loss of NINJ1 (which would also point at a more general antiviral mechanism). It is clear that all these mechanistic details cannot be resolved in a revision, but at the current stage, the observed difference is merely a potentially interesting observation that could not be traced down to any potential molecular model.

We thank the reviewer for their comments. We have now included data with VACV, another enveloped virus that uses endocytosis and micropinocytosis to enter cells, and we also observed higher levels of intracellular virions after a 10 min infection in cells lacking NINJ1 compared to WT cells. This suggests that NINJ1 affect entry of different viruses, not only HSV-1. This is now Figure 3F.

>The 'entry assay' is somewhat unconventional and is a little difficult to judge. For VACV 'entry', bound virions are not inactivated with citric acid, and it is therefore somewhat unclear what the assay really quantifies (it is also unclear if this treatment would inactivate VACV or induce its fusion with the plasma membrane). Perhaps it is easier to quantify VACV infection by flow cytometry (should be doable with the EGFP-expressing virus) and make more robust claims about other viruses.

>The text describes that 'Compared to WT cells, twice as many Ninj1^{-/-} cells were found to be infected with HSV-1'. This is not reflected in the quantification in 3G, in which infection on average is 31% and 45% in WT and Ninj1^{-/-} cells, respectively (according to the source data). Please correct the text.

Minor points: 1. Figure 2A: The data in figure 2A suggests that Pycard^{-/-} iBMDMs cannot be infected with HSV-1, although the authors do not comment on this. This needs to be explained, as it may also be interpreted as a potential risk that the different iBMDM cell lines may not be equally susceptible to infection due to differences during immortalization (assuming ASC does not play a role in the viral life cycle).

We thank the reviewer for pointing this out. Infection of Pycard^{-/-} iBMDMs was consistently low in our experiments but not always 0. We have replaced the previous figure panel with another more representative experiment (now Fig 2A,B).

>OK.

2. Figure 2E: Why was exclusively this experiment done with primary BMDMs? Please explain. Due to the weak signal of oligomers in the Nigericin sample, it cannot be ruled out that oligomers detection was just not sensitive enough. A more informative experiment would be to do the reconstitution experiments with NINJ1 mutants that cannot oligomerize/induce cell death (as described in Degen et al.) to evaluate which function of NINJ1 was required to restrict infection. Oligomerization of NINJ1 is an all or nothing readout. We either see oligomeric NINJ1 or not, giving us confidence that the weak signal is not masking a true oligomerization. We have also included additional replicates of this experiment in the source data, allowing readers to evaluate all of the data we collected.

>OK. I assume the authors can judge the outcome of NINJ1 oligomerization best and accept that the images are representative if no oligomerization is observed in three independent experiments (funny that the Nigericin response in repeat 2 is not that clear if the response is always 'all or nothing').

The reviewer also suggest we do the reconstitution experiments with NINJ1 mutants as we did in Degen et al. We attempted to perform this experiment but were not successful. The main reason was that these reconstitutions in primary cells relies on an mCherry-positive MSCV vector delivering mutant NINJ1, which has low infectivity, resulting in only 30-40% of macrophages cells actually expressing the protein. Subsequent infection with HSV-1 GFP yielded a very small fraction of double-positive cells, rendering the results uninterpretable. Repeating this experiment would be lengthy and challenging and our preliminary results did not suggest we would be successful.

>If the experiments in primary cells could not be done as intended, please at least comment why this experiment was done in primary cells and not the immortalized cell lines. This would help readers that would like to repeat NINJ1 oligomerization assays.

3. Figure 5A: While the text states that 'We observed a trend of a stronger induction of ISGs in the Ninj1^{-/-} cells with the most pronounced difference in Viperin expression and smaller differences in Bst2 and IFI2712a expression (Fig5a)', the actual data indicates a somewhat stronger transcription of Viperin in the WT than in the Ninj1^{-/-} cells (unless I misinterpret the data, in which case better explanations may be required). This needs to be clarified. Figure 5B merely shows (without any further statistics) that ISG transcription after HSV-1 infection relies on MyD88 or TRIF, as well as (partially) on STING. I am sure this has been investigated elsewhere in much more detail and depth and is likely not worth a main figure. It is not really assessed whether the stronger infection and subsequent ISG induction in Ninj1^{-/-} iBMDMs relies on these adaptors.

We thank the reviewer for this comment. The legend for this figure was switched between WT and NINJ1 and it is indeed the RNA level of ISGs in the NINJ1^{-/-} condition that are higher and this has now been fixed.

Indeed, the role of TLR and NFκB signaling in response to HSV-1 infection has been examined in other cell types, but never in macrophages. Thus, we are providing the first investigation of the sensing pathways employed by macrophages.

>OK.

4. Page 7: 'we infected sgLuciferase expressing control cells and CRISPR-generated NINJ1 KO cells.' I assume that cells expressing Cas9 and sgRNAs for luciferase or Ninj1 were compared, but the sentence is not very intuitive to understand. Please correct.

This has been clarified with text changes.

>OK.

Editorial side:

- Your manuscript will be published in our Reports section. Please combine the Results and Discussion sections and keep an eye on the character count, which should be approx. 27,000 characters incl. spaces but excluding methods and references.

Done, we are well below the 27,000 character limit.

- Please add the corresponding author's email address on the title page.

Done.

- Please add a callout for Fig. 4F in the text.

Done.

- Please upload the source data as one folder (zipped) per figure.

Done.

- Please add the reference to the source data images at BioStudies in the Data Availability section and please add a URL that resolves directly to the dataset.

Done.

- Materials and Methods should be Methods

Fixed.

- Please correct the labels on the EV figures to Expanded View Figure #.

These have been added.

- Also in the Expanded View Figure Legends the names need to be corrected to Figure EV# instead of Expanded View 1.

Done.

- Please define the error bars in the legends of figures 2D, E; 3G.

Done.

- Please specify the exact p values in the legend of figure 1A.

Done.

- Please indicate the statistical test used for data analysis in the legends of figures 1D, 3G, 1"

These have been added.

- Reagent table: please remove the Instructions paragraph.

Done.

- You cite two preprints, Ding X et al 2024 and Hartenian and Bernard 2025. The correct citation format for these is (preprint: Ding X et al, 2025) in the manuscript text. Addition of [PREPRINT] at the end of the reference in the reference list. For Hartenian and Bernard, 2025 I noted that it is not part of the reference list, it is only an in-text citation.

Thank you for pointing this out. This has been fixed as you suggested. One of the preprints has now been published and we have updated the reference list accordingly.

- Titles should not contain punctuation. What about: "NINJ1 blocks HSV-1 entry into macrophages to impact viral replication and immunity"

Thank you for the suggestion, we have changed it accordingly.

- Please write the abstract in present tense.

This has been fixed.

- Finally, EMBO Reports papers are accompanied online by

A) a short (1-2 sentences) summary of the findings and their significance

NINJ1 blocks HSV-1 entry into mouse macrophages independent of its role in cell death, allowing greater expression of pro inflammatory cytokines downstream of infection, controlling infection of these cells.

B) 2-3 bullet points highlighting key results

- Mouse macrophages lacking NINJ1 are more permissive to HSV-1 entry, resulting in higher levels of viral gene expression and virion production
- NINJ1 controls HSV-1 infection independently of its role in cell death
- Greater levels of HSV-1 replication in NINJ1-deficient cells leads to the release of fewer pro inflammatory cytokines

C) a schematic summary figure that provides a sketch of the major findings (not a data image).

Please provide the summary figure as a separate file in PNG or JPG format at a size of 550x300-600 pixels (width x height). Please note that the size is rather small and that text needs to be readable at the final size. Please send us this information along with the revised manuscript.

This is included in the resubmission.

Referee #1:

I find that the reviewers have addressed the points in a satisfactory manner, and that the conclusions are now fully supported by the data.

Thank you.

Referee #2:

The authors have adequately addressed the major comments from the reviewers as requested by the editor. They have replied to each of my specific minor comments as well.

Thank you.

Referee #3:

Hartenian et al. have revised their initial manuscript showing that ninjurin-1 (NINJ1), a macrophage membrane protein known to function in cell adhesion and cell death, restricts herpes simplex virus (HSV) at the entry stages of infection, apparently reducing the number of viral particles capable of entering murine macrophages. In the revised manuscript, they show that vaccinia virus is similarly restricted. The manuscript does not bring a mechanistic understanding of this restriction and will be of interest mostly to specialists.

In the response to initial review, authors have undertaken only modest revision, mostly to correct initial experimental shortcomings, but have not fundamentally increased mechanistic understanding or brought forward the relevance of their observations to HSV pathogenesis. The murine model of HSV is controversial but nevertheless has implicated both resident microglia (Reinhart et al., 2016 and 2021) as well as monocyte-derived macrophages (Katzilieri-Petras et al., 2022) in the brain in control of murine HSV encephalitis (significant work from the Paludan group that deserves better referencing and integration here). In addition, mouse monocyte-derived cell types are recruited to skin and become viral antigen positive following HSV infection (Eidsmo et al., 2009). Human monocyte-derived macrophages have also been a target of HSV study (Melchjorsen et al., 2006) and possibly control HSV along with virus-specific T cells in skin. Unfortunately, the relevance of murine bone marrow-derived macrophages to any setting of HSV infection or disease has not been established and the known complexities of HSV restriction in various mouse cells require further mechanistic insight and relevance in human cells.

Thank you for your comments, we have incorporated discussion of these studies in our manuscript.

Referee #4:

Hartenian et al. submitted a revised version to EMBO Reports, which addressed some of the reviewer's concerns and still describes an interesting finding that could not yet be mechanistically dissected. Overall, the manuscript meets the scope of EMBO Reports and can be published after minor revisions as pointed out below (my responses to the answers in the rebuttal letter are indicated with a '>').

Referee #1: Major points: 1. A major concern of this study is that the observed effect on HSV-1 infection is only shown in one cell line derived from Ninj1^{-/-} BMDMs that underwent spontaneous immortalization. It is difficult to compare WT and Ninj1^{-/-} iBMDMs as the mutations or cellular alterations that allow continuous division may be different between two cell lines (and no additional clones/independently generated immortalized lines were compared). Importantly, the authors show that inducible expression of NINJ1 reduces HSV-1 infection and therefore rescues restriction. This suggests that NINJ1-independent differences in the two cell lines do not account for the observed differences. However, as virus-mediated gene expression is used as the sole readout, it cannot be ruled out that strong overexpression of any gene affects GFP expression. This experiment would therefore need controls in which another irrelevant gene is overexpressed and GFP levels are not affected (we have observed that overexpression of a random gene alters expression of

virus-encoded reporters in our lab). Loss of NINJ1 did not affect infection in fibroblasts, ruling out any direct inhibitory effect of NINJ1 independent of the iBMDM context. It would therefore be more convincing that the role of NINJ1 is universal if the same phenotype was observed in WTE and KO primary BMDMs, independent KOs of NINJ1 derived from wt iBMDMs, or in cells in which NINJ1 was knocked down by RNAi (and in the best case other cell lines/models).

We thank the reviewer for their comments and for raising this issue.

We would like to emphasize that, in addition to using immortalized BMDMs, we also included data acquired in primary WT and NINJ1^{-/-} macrophages in our initial manuscript submission (see Fig. 1C). Importantly, we observed the same effect of NINJ1 deficiency, namely elevated viral replication, in both primary and immortalized cells. Therefore, the observed phenotype cannot be attributed to the process of immortalization.

To further strengthen this point, we have now included a panel showing a quantification of GFP by flow cytometry in these primary WT and NINJ1^{-/-} macrophages. This is now Figure 1D.

>Fig. 1C did not contain quantitative data and it is very subjective to judge which zoomed in area of the culture is representative. The new figure 1D answers the reviewer's concerns and shows that the described effect is also observed in primary cells that are not immortalized. The 'representative' microscopy images in 1C do not seem to show the same percentages of infected cells as the flow cytometry data in 1D. Please comment if any experimental conditions were different.

Thank you for your comment. Indeed, these experiments were done with different virus preps at more than 2 years of interval and while the relative trend is the same the absolute values are not. We have replaced the representative microscopy images in 1C with those with a prep of virus that is more consistent with what is used in 1D.

The reviewer is also concerned that the complementation in Figure 1H could be due to a more general effect of overexpressing a protein, instead of the complementation of the NINJ1-deficiency by exogenous expression of NINJ1.

This is a valid concern, and we have addressed it by expressing blasticidin resistance cassette lentivirally into the iNINJ1 cell line and comparing infection with and without expression of this transgene. We did not observe a change in replication kinetics, thus confirming that expression of an unrelated protein does not reduce HSV-1 replication. This is now Extended View Figure 1B.

>This reviewer point was addressed, although it would have been more accurate to use a lentiviral construct with the same promoter. If I understand it correctly, pLVX mNinj1 uses a dox-inducible promoter, while pLJM1 carries a CMV promoter. Please update the plasmid information in the reagent table to include both vectors.

Thank you for pointing this out, we have updated the reagent table accordingly.

2. The applied bulk plate reader measurements of GFP signal are convenient, but do not allow a clear distinction between the kinetics of viral gene expression and the fraction of infected cells. Figure 1A and B clearly show by flow cytometry that the fraction of infected cells is higher in Ninj1^{-/-} iBMDMs, while the level of EGFP expression is not clearly different. If more of the Ninj1^{-/-} iBMDMs are infected, it is to be expected that the different phases of gene expression are more sensitively detected by Western Blot (and no proof of sped up

kinetics). It is equally trivial that the infected culture in this case release more infectious progeny, that a larger fraction of cells exposes annexin V, and that a stronger ISG response is triggered. The interpretations therefore have to be specified, as the data does not support a reduction of 'all stages of the HSV-1 lifecycle'. I would further recommend using flow cytometry as a more definitive readout for infection for the most important experiments (which would also conveniently allow correlation with NINJ1 levels using antibody staining)

We thank the reviewer for pointing this out and have changed the text accordingly. We do note that by MFI the level of EGFP expression is increased (Figure 1B) which we now quantify and include as Extended View Figure 1A.

>Fig. EV1A shows a substantial change in one of the four quantified experiments, and I guess this can very much depend on the gates chosen for the quantification. In the shown representative curve, this is not too obvious. I would suggest rephrasing the sentence and toning down the conclusions on the MFI.

Thank you for this suggestion, we have toned down our conclusions related to the measured MFI.

3. Regarding the mechanism of action, it remains unclear what NINJ1 does, and whether its role is conserved beyond the observed differences in one cell line. This would be required to make the manuscript relevant enough for EMBO J. The authors find that more infectious virions can be recovered after endocytosis, indicating that there is enhanced uptake, or reduced degradation of virions in endosomes. They report a trend of more viral capsids in the cytosol, although it remains unclear if the accumulated data from three independent experiments would reveal the same trends if all three experiments were evaluated individually (no statistics are applied and strictly speaking, the assay does not distinguish bound and internalized intact virions from released cores).

Unless the authors show that HSV-1 is taken up by macropinocytosis, it is less telling that the fluid phase marker dextran is taken up similarly in wt and Ninj1^{-/-} iBMDMs. The easier comparison to evaluate similar and different endocytosis pathways would be to test if other viruses are also affected by the loss of NINJ1 (which would also point at a more general antiviral mechanism). It is clear that all these mechanistic details cannot be resolved in a revision, but at the current stage, the observed difference is merely a potentially interesting observation that could not be traced down to any potential molecular model.

We thank the reviewer for their comments. We have now included data with VACV, another enveloped virus that uses endocytosis and micropinocytosis to enter cells, and we also observed higher levels of intracellular virions after a 10 min infection in cells lacking NINJ1 compared to WT cells. This suggests that NINJ1 affect entry of different viruses, not only HSV-1. This is now Figure 3F.

>The 'entry assay' is somewhat unconventional and is a little difficult to judge. For VACV 'entry', bound virions are not inactivated with citric acid, and it is therefore somewhat unclear what the assay really quantifies (it is also unclear if this treatment would inactivate VACV or induce its fusion with the plasma membrane). Perhaps it is easier to quantify VACV infection by flow cytometry (should be doable with the EGFP-expressing virus) and make more robust claims about other viruses.

Thank you for your suggestion. Using citrate to inactivate extracellular virions is an assay that has been previously performed in the herpesvirus field to specifically measure entry (e.g. PMID 24453974, 33653890, 27630229). When we performed the control with VACV entry and citric acid, the citric acid was insufficient to inactivate VACV, which is consistent with literature showing low pH facilitates VACV entry (e.g. PMID 16940502) and as the reviewer

points out. Thus we did not include it in the figure in the revision. Instead, we washed stringently with PBS as we wished to capture the earliest stages of entry such that they would not be confounded by any additional potential effect of NINJ1 on gene expression.

>The text describes that 'Compared to WT cells, twice as many Ninj1^{-/-} cells were found to be infected with HSV-1'. This is not reflected in the quantification in 3G, in which infection on average is 31% and 45% in WT and Ninj1^{-/-} cells, respectively (according to the source data). Please correct the text.

Thank you, we have adjusted the text accordingly.

Minor points: 1. Figure 2A: The data in figure 2A suggests that Pycard^{-/-} iBMDMs cannot be infected with HSV-1, although the authors do not comment on this. This needs to be explained, as it may also be interpreted as a potential risk that the different iBMDM cell lines may not be equally susceptible to infection due to differences during immortalization (assuming ASC does not play a role in the viral life cycle).

We thank the reviewer for pointing this out. Infection of Pycard^{-/-} iBMDMs was consistently low in our experiments but not always 0. We have replaced the previous figure panel with another more representative experiment (now Fig 2A,B).

>OK.

2. Figure 2E: Why was exclusively this experiment done with primary BMDMs? Please explain. Due to the weak signal of oligomers in the Nigericin sample, it cannot be ruled out that oligomers detection was just not sensitive enough. A more informative experiment would be to do the reconstitution experiments with NINJ1 mutants that cannot oligomerize/induce cell death (as described in Degen et al.) to evaluate which function of NINJ1 was required to restrict infection.

Oligomerization of NINJ1 is an all or nothing readout. We either see oligomeric NINJ1 or not, giving us confidence that the weak signal is not masking a true oligomerization. We have also included additional replicates of this experiment in the source data, allowing readers to evaluate all of the data we collected.

>OK. I assume the authors can judge the outcome of NINJ1 oligomerization best and accept that the images are representative if no oligomerization is observed in three independent experiments (funny that the Nigericin response in repeat 2 is not that clear if the response is always 'all or nothing').

Thank you for your comment. By all or nothing we simply mean that if oligomerization has occurred it is visible on a gel irrespective of the intensity of the bands.

The reviewer also suggest we do the reconstitution experiments with NINJ1 mutants as we did in Degen et al. We attempted to perform this experiment but were not successful. The main reason was that these reconstitutions in primary cells relies on an mCherry-positive MSCV vector delivering mutant NINJ1, which has low infectivity, resulting in only 30-40% of macrophages cells actually expressing the protein. Subsequent infection with HSV-1 GFP yielded a very small fraction of double-positive cells, rendering the results uninterpretable. Repeating this experiment would be lengthy and challenging and our preliminary results did not suggest we would be successful.

>If the experiments in primary cells could not be done as intended, please at least comment why this experiment was done in primary cells and not the immortalized cell lines. This would help readers that would like to repeat NINJ1 oligomerization assays.

Thank you for your comment. In general we have had little success with reconstitution in the immortalized lines with different mutants as the variability of lentiviral transduction compounds differences in expression levels. Given that NINJ1 is known to auto activate upon high expression, having comparable levels of expression between mutants is quite important.

3. Figure 5A: While the text states that 'We observed a trend of a stronger induction of ISGs in the Ninj1^{-/-} cells with the most pronounced difference in Viperin expression and smaller differences in Bst2 and IFI2712a expression (Fig5a)', the actual data indicates a somewhat stronger transcription of Viperin in the WT than in the Ninj1^{-/-} cells (unless I misinterpret the data, in which case better explanations may be required). This needs to be clarified. Figure 5B merely shows (without any further statistics) that ISG transcription after HSV-1 infection relies on MyD88 or TRIF, as well as (partially) on STING. I am sure this has been investigated elsewhere in much more detail and depth and is likely not worth a main figure. It is not really assessed whether the stronger infection and subsequent ISG induction in Ninj1^{-/-} iBMDMs relies on these adaptors. We thank the reviewer for this comment. The legend for this figure was switched between WT and NINJ1 and it is indeed the RNA level of ISGs in the NINJ1^{-/-} condition that are higher and this has now been fixed.

Indeed, the role of TLR and NFKb signaling in response to HSV-1 infection has been examined in other cell types, but never in macrophages. Thus, we are providing the first investigation of the sensing pathways employed by macrophages.

>OK.

4. Page 7: 'we infected sgLuciferase expressing control cells and CRISPR-generated NINJ1 KO cells.' I assume that cells expressing Cas9 and sgRNAs for luciferase or Ninj1 were compared, but the sentence is not very intuitive to understand. Please correct. This has been clarified with text changes.

>OK.

Dear Prof. Broz,

Thank you for the submission of your further revised manuscript to our editorial offices. I now went through this and you p-b-p-response and I consider the remaining points of referees #3 and #4 as adequately addressed.

Before I can proceed with formal acceptance, I have these further editorial requests I ask you to address in a final revised manuscript:

- We will publish your manuscript in the report format. Scientific reports should have a main text of 25,000 (+/- 2,000) characters. Your manuscript presently has (without methods, references and legends) more than 32000. Please try to reduce the main text accordingly (to at least below 30000 characters). See also:

<https://www.embopress.org/page/journal/14693178/authorguide#researcharticleguide>

- Please provide titles for the EV figures in their legends.

- Please add molecular weight markers to all Western blots shown.

- Please provide statistical testing where applicable. Presently, no statistics has been done for panels 1G, 2B/C/D/E, 3A/J, 5A/B, all of EV3 and EV4B/C (it seems). Please add this. Please make sure that the number "n" for how many experiments were performed, their nature (biological versus technical replicates), the bars and error bars (e.g. SEM, SD) and the test used to calculate p-values is indicated in the respective figure legends (main and EV figures). Please also check that all the p-values are explained in the legend, and that these fit to those shown in the figure.

Please avoid the phrase 'independent experiment' that is still present in the legends (e.g. Figs. 5, EV3 and EV4) but clearly state if these were biological or technical replicates. Please also indicate (e.g. with n.s.) if testing was performed, but the differences are not significant. In case n=2, please show the data as separate datapoints without error bars and statistics. See also: <http://www.embopress.org/page/journal/14693178/authorguide#statisticalanalysis>

- Please add to each legend (main and EV figures, where applicable) a 'Data Information' section explaining the statistics used or providing information regarding replicates and scales. See:

- Please add the primer information directly to the reagents and tools table and remove the table from the methods section. Please add callouts to the R&T table.

- Please add a direct link (URL) to the data deposited at the BioImageArchive to the Data Availability Section.

- The scale bar(s) in panel 1C is (are) hardly visible. Please improve.

Best,

All editorial and formatting issues were resolved by the authors.

Prof. Petr Broz
University of Lausanne
Immunobiology
Chemin des Boveresses
Epalinges 1066
Switzerland

Dear Prof. Broz,

I am very pleased to accept your manuscript for publication in the next available issue of EMBO reports. Thank you for your contribution to our journal.

Yours sincerely,
